JCB Journal of Cell Biology

# Surface tension–driven sorting of human perilipins on lipid droplets

Ana Rita Dias Araújo[1]*, Abdoul Akim Bello[1]*, Joëlle Bigay[1]*, Céline Franckhauser[2]*, Romain Gautier[1], Julie Cazareth[1], Dávid Kovács[1], Frédéric Brau[1], Nicolas Fuggetta[2], Alenka Čopič[2], and Bruno Antonny[1]

**Perilipins (PLINs), the most abundant proteins on lipid droplets (LDs), display similar domain organization including amphipathic helices (AH). However, the five human PLINs bind different LDs, suggesting different modes of interaction. We established a minimal system whereby artificial LDs covered with defined polar lipids were transiently deformed to promote surface tension. Binding of purified PLIN3 and PLIN4 AH was strongly facilitated by tension but was poorly sensitive to phospholipid composition and to the presence of diacylglycerol. Accordingly, LD coverage by PLIN3 increased as phospholipid coverage decreased. In contrast, PLIN1 bound readily to LDs fully covered by phospholipids; PLIN2 showed an intermediate behavior between PLIN1 and PLIN3. In human adipocytes, PLIN3/4 were found in a soluble pool and relocated to LDs upon stimulation of fast triglyceride synthesis, whereas PLIN1 and PLIN2 localized to pre-existing LDs, consistent with the large difference in LD avidity observed in vitro. We conclude that the PLIN repertoire is adapted to handling LDs with different surface properties.**

## Introduction

Lipid droplets (LDs) differ markedly from other cellular organelles in that they are not surrounded by a phospholipid bilayer but by a monolayer that isolates the LD core made of triglycerides and other neutral lipids from the cytosol (Dhiman et al., 2020; Olarte et al., 2021; Henne, 2023). How this unique property of LDs contributes to the preferential targeting of some cytosolic proteins to LDs as compared with bilayer-bound organelles is a fundamental question, which has been addressed using biochemical reconstitution, cellular models, and molecular dynamic simulations (Bacle et al., 2017; Prévost et al., 2018; Čopič et al., 2018; Chorlay and Thiam, 2020; Kim and Swanson, 2020). The issue of protein targeting to LDs goes beyond a preference for monolayer versus bilayer-bound organelles. LDs are themselves heterogenous in size and composition (Wolins et al., 2006; Thul et al., 2017; Thiam and Beller, 2017). This diversity probably reflects the fact that LDs are associated with many different processes, including lipid storage, lipid channeling to other organelles, innate immunity, proteostasis, and cancer metabolism (Sánchez-Álvarez et al., 2022; Mathiowetz and Olzmann, 2024).

PLINs are highly abundant LD proteins that are often referred to as LD coats due to their ability to control the access of lipases and stabilize LDs (Greenberg et al., 1991; Blanchette-

Mackie et al., 1995; Brasaemle et al., 1997, 2004; Sztalryd and Brasaemle, 2017; Xu et al., 2019; Najt et al., 2022; Griseti et al., 2023). In humans, the PLIN family contains five members (PLIN1–5), which show notable differences in their tissue and subcellular distribution, mode of regulation, and protein interactions (Sztalryd and Brasaemle, 2017; Najt et al., 2022; Griseti et al., 2023). PLINs contain three main regions: (1) an N-terminal PAT domain, the structure of which is not known but might contain three helices (Choi et al., 2023), (2) a central disordered region, which has been shown to target LDs by providing 3–11 amphipathic helices (AH) that bind to the LD interface (Bussell and Eliezer, 2003; Bulankina et al., 2009; Rowe et al., 2016; Čopič et al., 2018; Giménez-Andrés et al., 2021), and (3) a C-terminal domain, whose structure has been solved in the case of PLIN3 as forming a four-helix bundle (Hickenbottom et al., 2004). However, PLINs show variations around this canonical scheme. For instance, the four-helix bundle region of PLIN1 contains hydrophobic segments that are required for LD targeting (Subramanian et al., 2004; Majchrzak et al., 2024) and is followed by a C-terminal extension critical for the control of lipolysis (Granneman et al., 2009; Gandotra et al., 2011). PLIN4 does not contain a PAT region, whereas its central AH is dramatically extended, containing 29 × 33-mer repeats (Čopič et al.,

[1]Université Côte d'Azur, CNRS and Inserm, Institut de Pharmacologie Moléculaire et Cellulaire, UMR 7275, Sophia Antipolis, France; [2]Centre de Recherche en Biologie Cellulaire de Montpellier-CRBM, Université de Montpellier, CNRS, UMR 5237, Montpellier, France.

*A.R. Dias Araújo, A.A. Bello, J. Bigay, and C. Franckhauser contributed equally to this paper. Correspondence to Joëlle Bigay: bigay@ipmc.cnrs.fr; Alenka Čopič: alenka.copic@crbm.cnrs.fr; Bruno Antonny: antonny@ipmc.cnrs.fr.

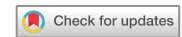

2018; Giménez-Andrés et al., 2021; Ruggieri et al., 2020). PLIN5 contains a C-terminal extension that allows it to tether LDs to mitochondria (Wang et al., 2011). Although progress has been made toward deciphering the various PLIN regions, the presence of intrinsically disordered regions and extended lipid-interacting surfaces, as well as the low solubility of some PLINs have hampered in-depth structural characterization. Therefore, our understanding of the mechanisms by which PLINs interact with the surface of LDs remains fragmental.

One long-standing question is why perilipins appear heterogeneously distributed on LDs. In 2D adipocyte cultures, there is a gradient of LD coverage by PLINs. PLIN1 outlines large and centrally localized LDs, PLIN2 is found at the surface of intermediate LDs, whereas PLIN3 and PLIN4 mark small and peripheral LDs (Wolins et al., 2003, 2005, 2006). This uneven distribution might result from differences in protein turnover during cell differentiation, and/or from intrinsic differences in the physicochemical properties of PLINs. These include the amino-acid composition of their AH regions, the presence of additional hydrophobic regions, or the ability of the PAT region or the four-helix bundle to undergo conformational changes (Ajjaji et al., 2019; Giménez-Andrés et al., 2021; Choi et al., 2023; Griseti et al., 2023). Because many proteins target LDs through AH, understanding the differential targeting of PLINs to LDs should have general implications.

The lipid composition of the LD surface and core (Chorlay and Thiam, 2020; Hsieh et al., 2012; Choi et al., 2023), as well as more general changes in LD physicochemical properties (Dhiman et al., 2020; Olarte et al., 2021; Henne, 2023; Griseti et al., 2023) could contribute to the differential binding of proteins. Notably, LD phospholipid coverage should vary according to the cellular metabolic state (e.g., neutral lipid synthesis versus lipolysis), resulting in differences in LD surface tension. Molecular dynamic simulations suggest that, at high surface tension, triolein molecules interdigitate with the phospholipid acyl chains or even participate in the monolayer by adopting an ordered conformation, thereby creating a lipid surface very different from that of a bilayer (Bacle et al., 2017; Kim and Swanson, 2020). Furthermore, the size and thus curvature of LDs varies over several orders of magnitude between different cells, and also within the same cell, for example in adipocytes during differentiation. In the case of bilayer-bound organelles, curvature in combination with conical lipids such as diacylglycerol acts as a targeting signal for proteins that detect lipid packing defects present in the bilayer (Bigay and Antonny, 2012; Vamparys et al., 2013; Vanni et al., 2014). Whether the packing defects present on small LDs, on LDs under tension, or on LDs undergoing changes in lipid composition resemble those present in lipid bilayers is not clear and difficult to tackle experimentally (Bacle et al., 2017; Prévost et al., 2018; Kim and Swanson, 2020; Čopič et al., 2018; Giménez-Andrés et al., 2021).

Directly addressing how PLINs differentially target LDs requires reconstitution experiments with full-length proteins, proper LD mimetics, and complementary assays. However, only PLIN2/3 have been purified in their full-length form (Najt et al., 2014; Sincock et al., 2003). Furthermore, artificial LDs for biochemical experiments are far from being mastered as well as artificial lipid bilayers, although significant progress has been made in recent years (Thiam et al., 2013a, 2013b; Wang et al., 2016; Julien et al., 2021; Titus et al., 2021; Gandhi et al., 2024). In this work, we combined immunofluorescence analysis of endogenous PLINs in human adipocytes with a new LD mimetic system amenable to transient tension challenge and to analytical assays, including flow cytometry for quantification of protein binding. We show that purified full-length PLIN1/2/3 display strikingly different LD binding properties, which are consistent with their different LD targeting in cells. As compared with changes in lipid composition, LD tension appears to be more decisive for PLIN differential binding to LDs, a result that may apply to other LD proteins.

## Results

### Analysis of endogenous PLINs in human adipocyte cultures suggests different modes of interaction with LDs

PLINs are present in all metazoans as well as in more distant species; however, their number has expanded in vertebrates and their sequences and tissue distribution in humans and other mammals suggest functional specialization (Griseti et al., 2023). We analyzed the distribution of endogenous PLINs in immortalized human adipocytes (TERT-hWA), which can be differentiated in cell culture dishes and endogenously express human PLIN1-4 (Markussen et al., 2017; Klingelhuber et al., 2024).

We applied an optimized differentiation protocol to these cells, which led to efficient differentiation of most cells in the culture dish over the course of 14 days, as shown by the progressive enlargement of LDs observed by transmission microscopy (Fig. 1 A), and the expression of the adipocyte marker hormone-sensitive lipase (HSL) (Fig. 1 B). We observed a large concomitant increase in the expression of adipocyte-specific PLIN1 and PLIN4, as shown by Western blot analysis using antibodies against endogenous proteins (Fig. 1 B) in accordance with previous observations (Wolins et al., 2003). PLIN2 could be detected already in non-differentiated cells and showed a modest increase during differentiation, whereas PLIN3, which is widely-expressed in non-differentiated cells, remained about constant (Fig. 1 B), as reported (Barneda and Christian, 2017).

We then analyzed the localization of different PLINs by immunofluorescence (IF) followed by 3D confocal imaging. For this analysis, we selected three different time points of the differentiation protocol, early (day 3), intermediate (day 6), and late (day 10), correlating with the largest increase in LD size. We assessed the localization of PLIN1, PLIN2, and PLIN4 in the same cells under two conditions: in normal growth media or after the addition of 100 μM oleic acid (complexed with BSA; OA) for 2.5 h before fixation and IF analysis (Fig. 1 C; and Fig. S1, A and B). Under all conditions, we observed PLIN1 on centrally localized LDs whose size progressively increased with differentiation, but did not show a large change after OA treatment. In striking contrast, PLIN4 displayed diffuse cytosolic signal in almost all non-treated cells at all stages of differentiation, with only faint staining of some peripheral LDs in a fraction of cells. Upon addition of OA, PLIN4 relocalized to peripheral LDs in a large fraction of cells (50% or more), with only a low number of cells

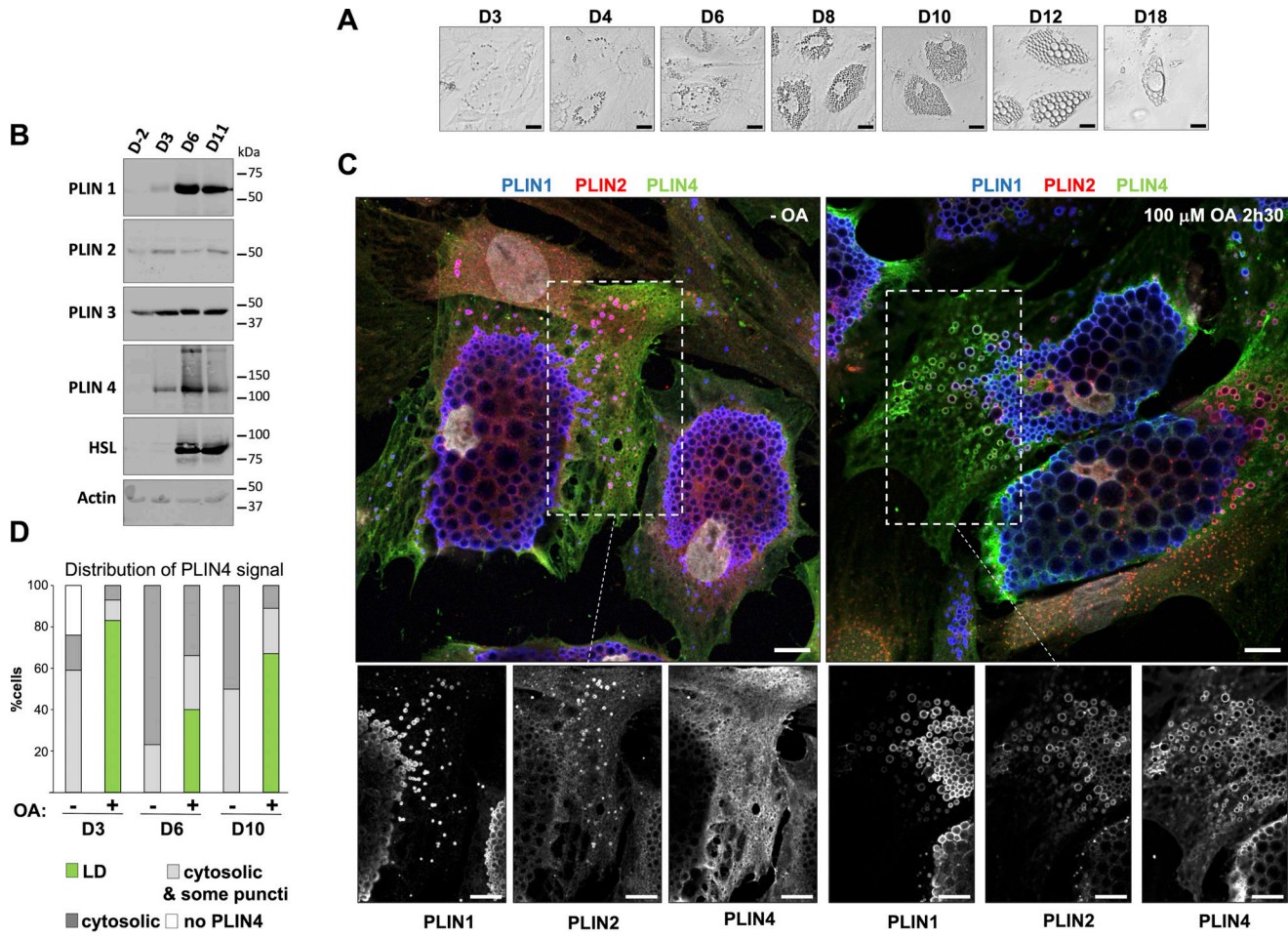

Figure 1. **Analysis of endogenous PLINs in human adipocyte (TERT-hWA) cultures suggests different modes of interaction with LDs. (A)** Time course of adipocyte differentiation. **(B)** Western blot analysis of adipocyte markers and PLINs during adipocyte differentiation. P: proliferation; D3, D6, and D11: day of differentiation. **(C)** Representative z sections obtained by confocal fluorescence microscopy of endogenous PLIN1 (blue), PLIN2 (red), and PLIN4 (green) in human adipocytes at day 10 of differentiation after immunofluorescence with specific antibodies, with the nuclei stained with DAPI (white). Bottom panels show the three protein channels in the indicated area. The cells were maintained in culture medium or fed with 100 µM oleic acid for 2.5 h before observation. Scale bar: 10 µm. **(D)** Quantification showing the fraction of cells with PLIN4 signal as indicated under the same conditions as in C. Four categories were defined: cells without detectable PLIN4; cells in which PLIN4 was entirely cytosolic; cells in which PLIN4 was mostly cytosolic but marked a few puncti (possible LDs); cells in which PLIN4 was LD-localized. N of cells quantified was 108 for D3 − OA, 104 for D3 + OA, 92 for D6 − OA, 92 for D6 + OA, 98 for D10 − OA and 93 for D10 + OA, from one of two representative experiments. See Fig. S1, B and C, for representative immunofluorescence images from D3 and D6 of adipocyte differentiation. Source data are available for this figure: SourceData F1.

retaining only cytosolic signal (Fig. 1, C and D). Importantly, this effect was observed both early and late in differentiation. There was some heterogeneity in the differentiating adipocyte population, and we observed that, on the same day, in the cells with less abundant or smaller PLIN1-labeled LDs, PLIN4 more frequently relocalized to small LDs after OA addition. PLIN2 displayed an intermediate behavior: it could be observed in the cytosol and on more peripheral/smaller LDs in the absence of OA, with a slight increase after OA addition, but the result also depended on the stage of differentiation and should be further evaluated (Fig. 1 C; and Fig. S1, A and B). We also analyzed the localization of PLIN3 at day 6 and observed behavior very similar to PLIN4, with cytosolic PLIN3 signal in the absence of OA, and an even higher LD localization after OA addition (Fig. S1 D).

These results are in agreement with previous observations in mouse-differentiated 3T3-L1 adipocyte cells that showed relocalization of PLIN4 and PLIN3 to small peripheral LDs after OA addition, leading to the conclusion that these PLINs specifically localize to nascent LDs (Wolins et al., 2003, 2005). However, our analysis at different stages of adipocyte differentiation instead suggests that PLIN4 and PLIN3 function as buffers, localizing to newly formed LDs after a burst in the production of triglycerides that are induced by OA addition, but they do not bind to LDs that form gradually during adipocyte differentiation.

**Rationale for LD preparation and protein binding protocols**

To understand the basis for the differences in PLIN recruitment to cellular LDs, we wished to set up an LD mimetic system that would allow us to study the interaction of purified full-length PLINs with artificial LDs. Such LDs should be of defined composition and size and should be amenable to complementary assays including visualization by fluorescence microscopy, bulk

biochemical assays, and flow cytometry analysis to gather quantitative data from all individual LDs in a mixture. However, manipulating LDs in vitro is challenging. First, the low oil density (≈0.9 g/ml) makes LDs light objects that readily float in a microscopy chamber or in a test tube, impeding homogenous contact with the bulk phase. Second, the gentle addition of a protein to a preformed oil droplet covered with phospholipids might lead to a false negative result (i.e., no binding) because the phospholipid surface is too dense to accommodate the protein, whereas the protein may be well adapted to LDs with a lower phospholipid coverage. The last consideration might be important for PLINs, which are the most abundant LD-associated proteins and contain long (a hundred aa) or very long (a thousand aa) AH motifs (Brasaemle et al., 2004; Xu et al., 2019; Čopič et al., 2018; Hynson et al., 2012). In addition, cellular data (Fig. 1) suggest that some PLINs preferentially bind to quickly forming LDs, i.e., under conditions where the phospholipid/neutral lipid ratio might not be at equilibrium. Therefore, it should be informative to perform LD binding experiments not only with stable preformed LDs but also with LDs experiencing transient tension stress.

We established a new LD binding assay that allowed us to both control the LD density and apply a deformation step to transiently increase surface tension (Fig. 2 A). The LD core was made of a mixture of triolein (TG[18:1/18:1/18:1]) and heavy oil that is denser than water (brominated vegetable oil, BVO). By adjusting the ratio of the two oils, one can prepare artificial droplets with a defined density (Julien et al., 2021). We chose a density of 1.05 g/ml, which corresponded to a brominated oil: triolein ratio of 1:2. Thus, triolein dominated the oil phase but the droplets had a slightly higher density than the buffer, which enabled sedimentation of LDs in an optical chamber. Phospholipids were added at a phospholipid/oil ratio that gave stable LDs with a defined average diameter, which were further sized by an extrusion step (see Materials and methods). We chose an average LD diameter of 2 μm for flotation and flow cytometry experiments, and 10 μm for light microscopy experiments. This diameter range is similar to that of many cellular LDs.

We compared protein binding to these artificial LDs (aLD) under two conditions: when proteins were gently added to aLDs and when we applied a second extrusion step to the aLD–protein mixture (Fig. 2 A). In this second step, we forced the aLD–protein mix to pass through a polycarbonate filter of a smaller pore size than the average diameter of the initial aLD suspension (e.g., 1 μm versus 2 μm or 8 μm versus 10 μm). This should create a transient tension at the aLD interface by deforming the initially spherical aLDs. The resulting increase in the surface-to-volume ratio might facilitate protein insertion.

### PLIN3 binding to aLDs is hypersensitive to surface tension

We first performed experiments with full-length PLIN3 and with phosphatidylcholine (PC)-covered aLDs (Fig. 2 B). PLIN3 is intrinsically soluble and its purification does not require any detergent or chaotropic agent (Hynson et al., 2012; Sincock et al., 2003; Choi et al., 2023). PC is the most abundant phospholipid found on LDs in cells (Chitraju et al., 2012; Bartz et al., 2007). To facilitate the analysis, PLIN3 was labeled with AlexaFluor488

(AF488) and used at 5–20 mol% compared with unlabeled PLIN3, whereas the phospholipids included a fraction of 16:0 lissamine rhodamine B phosphatidylethanolamine (Rho-PE) as a fluorescent tracer. Because acyl chain unsaturation can strongly affect the interaction of AH-containing proteins with lipid membranes (Vanni et al., 2014), we used PC species with increasing levels of unsaturation, namely PC(14:0/14:0) (fully saturated), PC(16:0/18:1) (saturated-monounsaturated), and PC(18:1/18:1) (di-monounsaturated).

We incubated purified PLIN3 for 10 min with aLDs, which were subsequently isolated by flotation on sucrose cushions. Under these conditions, only a small fraction of PLIN3 was recovered in the top fraction containing aLDs. We observed slightly more binding to PC(18:1/18:1)-covered aLDs (≈10%) than to PC(14:0/14:0)- or PC(16:0/18:1)-covered aLDs (≈5%) (Fig. 2 B). In contrast, the level of PLIN3 recovery in the aLD fraction dramatically increased upon extrusion, reaching values close to 50% regardless of the PC species (Fig. 2 B). This increase was observed for both AF488 PLIN3, as directly visualized by SDS-PAGE before staining, and for total PLIN3, as detected after total protein staining (Fig. S2 A), suggesting that fluorescent labeling did not modify the affinity of the protein for aLDs. High-performance thin layer chromatography (HPTLC) (Fig. S2 B) and SDS-PAGE (Fig. 2 B) showed that the majority (≈75%) of lipids and protein was recovered after extrusion.

We next conducted flow cytometry measurements on similar aLD–protein mixtures. Plotting the side (SSC) versus forward light scattering signals (FSC) of the various aLD–protein suspensions showed an S shape, which is characteristic of well-defined lipid emulsions (Fig. S2 C) (Fattaccioli et al., 2009). Analyzing the Rho-PE channel to follow droplets and the AF488 channel for PLIN3 before extrusion showed that only a small fraction of the droplets exhibited the protein signal (≈10%). After extrusion, the large majority of aLDs (≈90%) displayed the two signals, confirming the dramatic effect of extrusion on PLIN3 recruitment to the aLD surface (Fig. 2 C).

Next, we performed several complementary experiments. First, we increased the relative amount of brominated oil at the expense of triolein in the aLD formulation and performed experiments with both small (2 μm) and large (10 μm) aLDs. The binding of PLIN3 to the aLDs did not change upon brominated oil increase (0, 33, 67, or 100%) and remained strongly dependent on extrusion (Fig. S2 D), suggesting that the presence of brominated oil did not modify the surface properties of the aLDs. The effect of extrusion was observed for both small (2 μm) and large (10 μm) aLDs although the latter showed more variability in PLIN3 binding (Fig. S2 D). Large aLDs might be more fragile objects to manipulate. Second, we varied the ratio between labeled and unlabeled PLIN3 and obtained similar results, confirming that PLIN3 labeling did not modify its binding properties (Fig. S2 E). Third, we performed aLD binding experiments with a fragment of PLIN4 that encompasses a large part of its giant AH region (aa 510–905, PLIN4 AH-12mer) and can directly emulsify triolein (Čopič et al., 2018; Giménez-Andrés et al., 2021). This fragment behaved similarly to PLIN3 in the aLD-binding assay, showing almost no spontaneous binding to PC-covered aLDs and strong binding after extrusion (Fig. S2 F).

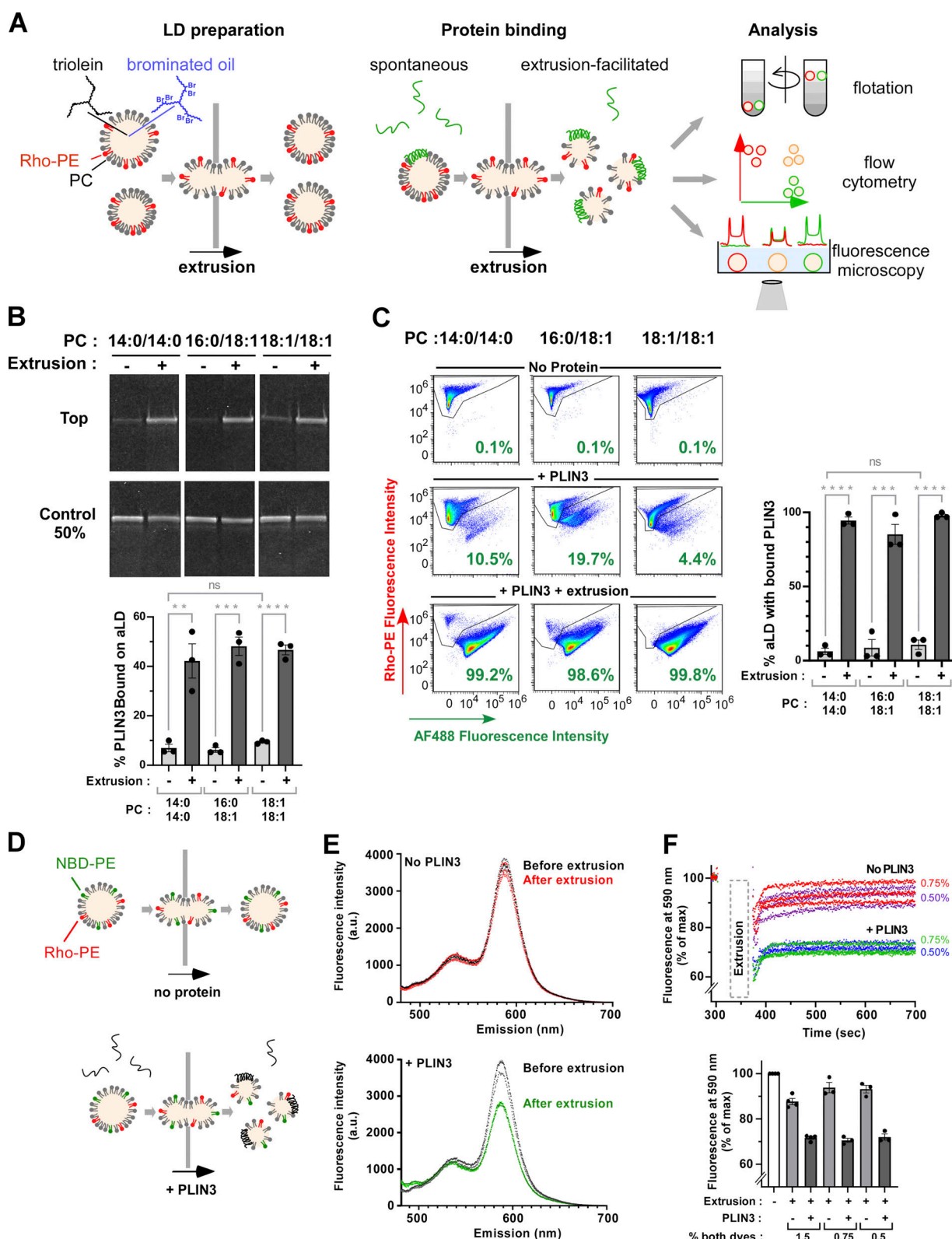

**Figure 2.** **Binding of PLIN3 to aLDs requires surface tension. (A)** Rationale of the aLD binding assay. PC-covered aLDs containing a mix of heavy (bro-minated) and normal (triolein) oil were sized by a first extrusion step. Thereafter, a protein of interest was gently added. Half of the sample was processed directly for analysis. The other half was submitted to a second extrusion step to transiently increase aLD surface tension. Three complementary methods were used for analysis: flotation to determine the fraction of aLD-bound protein; flow cytometry to determine the fraction of aLDs with bound protein; fluorescence microscopy to determine the surface coverage by proteins versus phospholipids. **(B)** Flotation analysis. 1.4 µM PLIN3 (of which 5% was labeled with AF488) was mixed with PC-covered aLDs (volume fraction of oil = 0.75%; calculated diameter = 2 µm; concentration of phospholipid [PL] = 62.5 µM). When indicated, the mixture was further extruded through 1-µm polycarbonate filters. After flotation, the top fraction was analyzed by SDS-PAGE and direct AF488

fluorescence detection. Bar plot: quantification (mean ± SEM) of the fraction of aLD-bound AF488 PLIN3 ($N = 3$). **(C)** Flow cytometry analysis of PLIN3 + aLD samples similar to those used in B. Bar plot: quantification (mean ± SEM) of the fraction of AF488 PLIN3-positive aLDs ($N = 3$). P values in B and C were calculated using an unpaired $t$ test. n.s., P > 0.05, **P < 0.01, ***P < 0.001, ****P < 0.0001. **(D)** Schematic view of the FRET assay to monitor the impact of extrusion on aLD phospholipid coverage. **(E)** Emission spectra of PC(16:0/18:1)-covered aLDs (2 μm) containing 0.75 mol%, NBD-PE and 0.75 mol% Rho-PE and supplemented (bottom) or not (top) with PLIN3. Spectra were recorded before (black) or after (colored traces) extrusion. NBD-PE was excited at 455 nm. Three independent measurements are shown for each condition. **(F)** Same as in E but in the kinetic mode: Rho-PE emission was continuously recorded at 590 nm upon excitation at 455 nm. At $t = 300$ s, the aLD suspension was removed, extruded, and put back immediately in the fluorimeter ($t = 375$ s). The lower bar plot summarizes the results from three independent extrusions and with 0.5, 0.75, or 1.5 mol% fluorescent lipids. Source data are available for this figure: SourceData F2.

Finally, we aimed to directly assess the effect of extrusion on the coverage of aLDs by phospholipids. For this, we used fluorescence resonance energy transfer (FRET) between two fluorescent phospholipids. By deforming the aLDs and increasing their surface, extrusion should spread apart the fluorescent probes, thereby decreasing FRET efficiency (Fig. 2 D). We chose NBD-PE (1,2-dioleoyl-sn-glycero-3-phosphoethanolamine-N-[7-nitro-2-1,3-benzoxadiazol-4-yl]) and Rho-PE, a donor/acceptor pair that has been extensively used to monitor liposome fusion (François-Martin and Pincet, 2017).

We prepared 2 μm POPC-covered aLDs containing 0.25–1.5 mol% of both NBD-PE and Rho-PE, a range of surface concentration at which the relative emission spectrum of NBD-PE and Rho-PE varied strongly due to FRET (Fig. S2, G and H) (François-Martin and Pincet, 2017). When we extruded the aLD suspension through a 1-μm filter and then recorded the fluorescence signal, the Rho-PE signal was lower than that observed before extrusion but rapidly recovered with an apparent half-time of 10 ± 5 s ($N = 6$) (Fig. 2, E and F). This observation indicated that extrusion caused transient spreading of the phospholipid monolayer. When we performed the same experiment in the presence of unlabeled PLIN3, a large FRET decrease was also observed, but it was followed by a much lower recovery (Fig. 2, E and F), suggesting that PLIN3 binding maintained the phospholipids apart (Fig. 2 D). By calibrating the FRET signal using aLDs covered with a decreasing percentage of the two fluorescent phospholipids (Fig. S2 H), we estimated that extrusion transiently spread apart the phospholipids by about twofold. Overall, these observations validate the extrusion strategy and indicate that binding of PLIN3 to aLDs is strongly favored by a transient decrease in phospholipid coverage, that is, an increase in surface tension.

### Surface tension is a more decisive parameter for PLIN3 binding than LD monolayer composition

Next, we performed PLIN3 binding experiments with aLDs displaying a more complex lipid composition. Specifically, we tested the effect of 1,2 dioleoyl-glycerol (DAG[18:1/18:1]) and used aLDs, which were covered by a phospholipid mixture very close to that revealed by lipidomic measurements (Chitraju et al., 2012; Bartz et al., 2007).

Diacylglycerols are intermediates in LD metabolism and have been shown to promote the binding of PLIN3 to lipid bilayers (Skinner et al., 2009; Ben M'barek et al., 2017; Choi et al., 2023; Stribny and Schneiter, 2023). In addition, DAG(18:1/18:1) increases the propensity of triolein to phase-separate in bilayers to form nascent LDs (Zoni et al., 2021). We first quantified the

effect of DAG(18:1/18:1) on PLIN3 binding to bilayers by using the membrane environment-sensitive fluorescent probe NBD (Čopič et al., 2018). We incubated NBD-labeled PLIN3 with liposomes of defined lipid composition and used NBD fluorescence intensity as an index of protein binding (Fig. 3, A and B). NBD–PLIN3 did not bind to PC(14:0/14:0) or PC(16:0/18:1) liposomes and showed modest binding to PC(18:1/18:1) liposomes, whereas strong binding was observed on diphytanoyl (4Me) PC liposomes (Fig. 3 A). The branched acyl chains of PC(4Me) promote large lipid packing defects in membranes, which facilitate the binding of AHs, including the long AH of PLIN4 (Garten et al., 2015; Čopič et al., 2018). Increasing the amount of DAG(18:1/18:1) at the expense of PC(16:0/18:1) led to a modest increase in PLIN3 binding to liposomes, which became significant only at DAG(18:1/18:1) levels >10 mol% (Fig. 3 B). We then asked whether DAG(18:1/18:1) could alleviate the need for tension in the case of PLIN3 binding to aLDs. Surprisingly, increasing the amount DAG(18:1/18:1) in the aLD formulation had no effect on the binding of PLIN3, which remained highly dependent on extrusion (Fig. 3 C).

Next, we performed experiments using aLDs covered by a phospholipid mixture inspired by lipidomic studies, namely 68% PC, 25% phosphatidylethanolamine (PE), and 7% phosphatidyl-inositol (PI) (Chitraju et al., 2012; Bartz et al., 2007). Spontaneous binding of PLIN3 to such aLDs was not improved as compared PC-covered aLDs (Fig. 3 D). Furthermore, including 20 mol% DAG(18:1/18:1) in the formulation did not increase PLIN3 binding and, under all conditions tested, we observed a dramatic increase in PLIN3 binding upon extrusion (Fig. 3 D). We conclude that PLIN3 does not bind spontaneously to aLDs even when they are composed of a lipid monolayer that mimicks authentic LDs or enriched in DAG(18:1/18:1), whereas it binds avidly to aLDs under tension.

### Systematic LD-binding measurements indicate a very large affinity range among PLINs

We next wanted to evaluate the binding of other PLINs to aLDs. However, PLIN1 and PLIN2 were more challenging to express and purify than PLIN3. Notably, PLIN1 and PLIN2 were insoluble in standard buffers. To overcome these difficulties, we set up different expression and purification procedures (Fig. S3 A). PLIN1 was expressed at low temperature in *E. coli* arctic cells. The solubility of PLIN1 and PLIN2 was improved by the addition of urea (Fig. S3 A).

We determined the minimal concentration of urea that kept the various PLINs soluble without impacting their binding to aLDs. Three observations suggested that 2 M urea was an

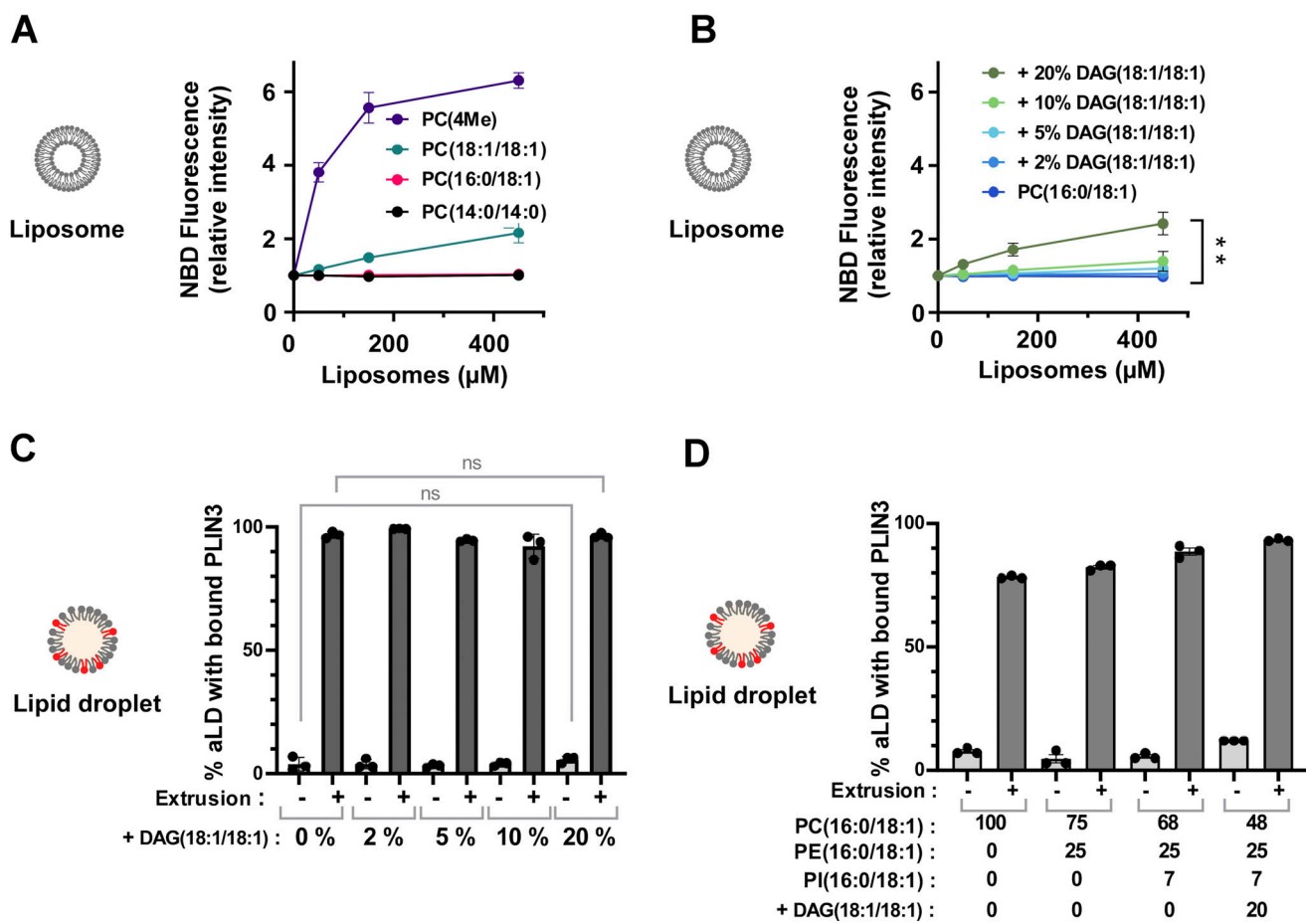

Figure 3. **Comparison of the interaction of PLIN3 with liposomes and aLDs prepared with various phospholipid mixes and diacylglycerol. (A and B)** Liposome-binding assays showing the fluorescence (mean ± SEM, $N = 3$) of NBD PLIN3 in the presence of increasing amounts of liposomes. Experiments were performed with PC liposomes of defined acyl chain composition (A) or with PC(16:0/18:1) liposomes containing increasing % of DAG(18:1/18:1) (B). DAG(18:1/18:1) slightly favors the binding of NBD PLIN3 to PC liposomes. The means of conditions 0 and 20% DAG were compared using an unpaired $t$ test. P value = ** < 0.01. **(C)** Flow cytometry analysis of AF488 PLIN3 binding to aLDs (mean of % ± SEM, $N = 3$) in the presence of increasing % of DAG(18:1/18:1). n.s., P > 0.05 according to unpaired $t$ test performed on the indicated conditions. **(D)** Flow cytometry analysis of AF488 PLIN3 binding to aLDs (mean of % ± SEM, $N = 3$). aLDs were prepared with different phospholipid ratios and with or without DAG(18:1/18:1). For panels C and D, the experimental conditions were as in Fig. 2, B and C.

optimal concentration. First, 2 M urea was sufficient to keep both PLIN1 and PLIN2 soluble (Fig. S3 B). Second, binding of PLIN3 to aLDs was not affected by the presence of 2 M urea and remained highly facilitated by extrusion (Fig. S3 C). Third, urea at 2 M did not modify the pattern and the kinetics of limited proteolysis of PLIN1, PLIN2, and PLIN3 by subtilisin. At both 0 and 2 M urea, PLIN3 showed a proteolysis-resistant C-terminal domain, corresponding to the C-terminal four-helix bundle; PLIN2 showed a similar profile but with less resistant fragments; PLIN1 was susceptible to proteolysis along its entire length (Fig. 4 A and Fig. S3 D). These observations suggest that PLIN1, PLIN2, and PLIN3 differ in the balance between ordered and disordered regions, and that their overall domain organization is not affected by 2 M urea.

Fig. 4 B shows aLD binding experiments with full-length PLIN1, PLIN2, or PLIN3 as analyzed by flow cytometry. PLIN1 bound spontaneously to aLDs regardless of the PC species (14:0/14:0, 16:0/18:1, or 18:1/18:1) at the surface. However, the AF488 signal was less intense on the PC(14:0/14:0)-covered aLDs,

suggesting lower protein coverage (Fig. 4 C). For PLIN2, we observed a large effect of PC unsaturation, with the fraction of PLIN2-positive aLDs increasing from 10% with PC(14:0/14:0)-covered aLDs up to 100% with PC(18:1/18:1)-covered aLDs (Fig. 4, B and C). Because of the low spontaneous binding of PLIN2 to aLDs covered with saturated PC, we aimed to test the effect of extrusion. However, we noticed that part of the aLD-PLIN1 and aLD-PLIN2 particles might have aggregated because the light scattering diagrams did not show a well-defined S-shape (Fig. S3 E). Excess protein in the case of PLIN1 and PLIN2 might be less tolerated compared to PLIN3, which is very soluble. We thus repeated these experiments with a fourfold lower protein concentration and observed very similar results in terms of percentage of PLIN-positive aLDs, although we noticed a lower AF488 signal suggesting less bound-protein, as expected (compare Fig. 4, B–E). We then applied extrusion to the PLIN2-aLD samples and observed that all aLDs became PLIN2 positive (Fig. 4, D and E), suggesting that PLIN2 is also sensitive to LD surface tension but that this requirement can be bypassed when

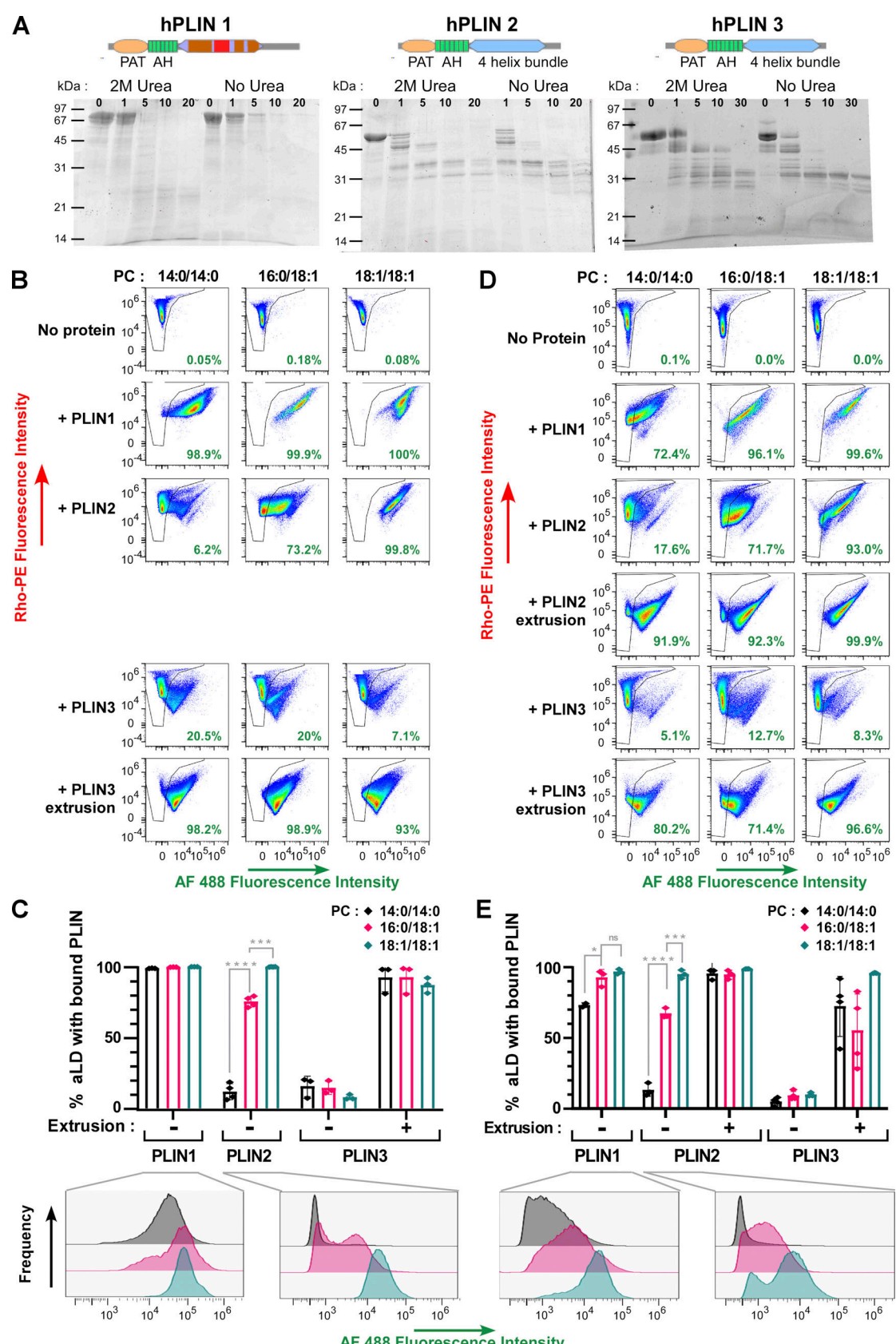

Figure 4. **Comparison of full-length PLIN1, PLIN2, and PLIN3 binding to aLDs. (A)** Predicted domain organization of PLIN1/2/3 and time-course of limited proteolysis in the presence of subtilisin and in the presence or absence of 2 M urea. Proteins were analyzed by Sypro orange staining. For a more complete view see Fig. S3 D. **(B–E)** Flow cytometry analysis of PLIN1, PLIN2, or PLIN3 in the presence of PC-covered aLDs (2 μm). When indicated, protein binding was

analyzed before and after extrusion. The protein concentration was 1.4 µM (B and C) and 0.35 µM (D and E). Bar plots in C and E show the quantification (mean ± SEM) of the fraction of AF488 PLIN positive aLDs from experiments similar to those shown in A and C, respectively (n = 3). P values in B and C were obtained with an unpaired t test. *P < 0.05, ***P < 0.001, ****P < 0.0001. For some conditions, the AF488 intensity distribution is shown to illustrate differences in PLIN coverage on the PLIN-positive aLDs.

the PC monolayer is sufficiently unsaturated. Altogether, these experiments indicate that the three PLINs have very different affinities for PC-covered aLDs, with the order PLIN3 << PLIN2 << PLIN1. We note that this order matches their calculated hydrophobicity (Giménez-Andrés et al., 2021; Ajjaji et al., 2019) and experimentally measured solubility in the presence or absence of urea (Fig. S3 B).

## Visualization of PLIN3/4 binding to LDs reveals phospholipid exclusion

From the flow cytometry analysis (see Fig. 2 C), we noticed that PLIN3 binding to aLDs was accompanied by a large (5–10-fold) decrease in the Rho-PE signal, suggesting that PLIN3 binding correlated with a reduction in aLD size and/or phospholipid coverage. A similar trend was observed for PLIN2 (Fig. 4, B and D). We thus determined the aLD size by dynamic light scattering measurements as well as by directly visualizing the aLDs in a flow cytometry apparatus equipped with a camera. Whereas the mere addition of PLIN3 did not modify aLD size, the combination of PLIN3 addition and LD extrusion led to a significant reduction in aLD diameter (Fig. S4, A and B). In agreement with the FRET data (see Fig. 2, D–F), this observation suggests that, in addition to phospholipids, PLIN3 makes a significant contribution to the surface of aLDs, hence enabling the formation of smaller aLDs.

To directly visualize the protein and phospholipid coverage on the aLDs, we used large PC(16:0/18:1)-covered aLDs, which could be observed by light microscopy. Fig. 5 A compares typical microscopy fields of aLDs supplemented with PLIN3 before and after extrusion. In the absence of protein, all aLDs were delimited by a similar phospholipid signal, as visualized in the Rho-PE channel. Upon gentle addition of PLIN3, about one-third of the aLDs appeared covered by AF488 PLIN3, whereas the remaining aLDs seemed devoid of it. Interestingly, the PLIN3-positive aLDs exhibited a weak Rho-PE signal as compared to the PLIN3-negative droplets. After extrusion, we could no longer distinguish different aLD populations, but we rather observed a continuum of aLDs displaying various levels of AF488 and Rho-PE fluorescence on their contour. To quantify the relative coverage of the aLDs by PLIN3 and phospholipids, we determined the Rho-PE and AF488 fluorescence profiles of hundreds of aLDs and ranked them according to their coverage by AF488 PLIN3 (Fig. 5 A). The analysis confirmed the presence of two aLD populations before extrusion. The first population (≈65% of LDs) displayed a phospholipid coverage similar to the initial aLD preparation and a PLIN3 signal that did not exceed 10% of the signal of the protein in solution. The second aLD population (≈35%) displayed a clear PLIN3 signal, but with a lower Rho-PE signal compared with the PLIN3-negative aLDs or to the initial aLD preparation. For this second population, the Rho-PE signal and the AF488 signal correlated in an inverse manner; at maximal PLIN3 intensity, the Rho-PE intensity dropped to ≈20% of

the initial signal (Fig. 5, A and B). After extrusion, almost all aLDs displayed AF488 PLIN3 whereas the Rho-PE signal was 3.5 times lower than that initially (Fig. 5 B). This analysis suggests that PLIN3 binding to aLDs correlates with a strong reduction in phospholipid coverage.

We performed a similar analysis with the AH region of PLIN4, for which previous studies demonstrated that it could directly coat and emulsify triolein in the absence of phospholipids and that it could rescue the size of LDs in cells after PC depletion (Čopič et al., 2018; Giménez-Andrés et al., 2021). PLIN4 AH 12mer showed an even higher sensitivity to extrusion than PLIN3. Before extrusion, only a few percent of aLDs showed a detectable AF488 PLIN4 AH signal. After extrusion about 25% aLDs became 488 PLIN4 AH positive. Furthermore, we observed a strong anticorrelation between the AF488 PLIN4 AH and the Rho-PE signals (Fig. 5 C).

## Recruitment of PLIN4 to LDs in adipocytes depends on the rate of LD formation

Given the striking sensitivity of PLIN4 AH to aLD tension in vitro (Fig. 5 C), we surmised that the redistribution of endogenous PLIN4 to newly formed LDs in TERT-hWA cells (see Fig. 1) might be sensitive to the rate at which LDs form. When cells synthesize triglycerides at a speed that is too fast to be accompany by phospholipid coverage, this should increase surface tension of the newly formed LDs, thereby favoring PLIN4 recruitment. In contrast, when triglyceride synthesis is slower, this should be unfavorable for PLIN4 recruitment as phospholipid synthesis might be sufficient to promote LD coverage, hence preventing the increase in surface tension. To test this hypothesis, we compared the effect of adding oleic acid either as a single burst of 100 µM or as five additions of 20 µM every half an hour (Fig. 6 A). Whereas the first protocol promoted the efficient recruitment of endogenous PLIN4 to LDs in most differentiated TERT-hWA cells, the second protocol was less efficient (Fig. 6, B and C), indicating that PLIN4 binds preferentially to fast forming LDs.

## Visualization of PLIN3 binding to bilayers does not reveal phospholipid exclusion

The cellular and biochemical experiments presented so far support a model in which the soluble perilipins PLIN3 and PLIN4 sense the surface tension of newly formed LDs and bind to them when there is a deficit in phospholipid coverage. These perilipins act as substitute of phospholipids, hence the link between PLIN3/4 binding to aLDs and low phospholipid coverage.

However, PLIN3 and PLIN4 have been shown to also target lipid bilayers under some circumstances, both in vitro (e.g., binding to diphytanoyl liposomes) and in vivo (e.g., binding to DAG-enriched ER or the plasma membrane) (Stribny and

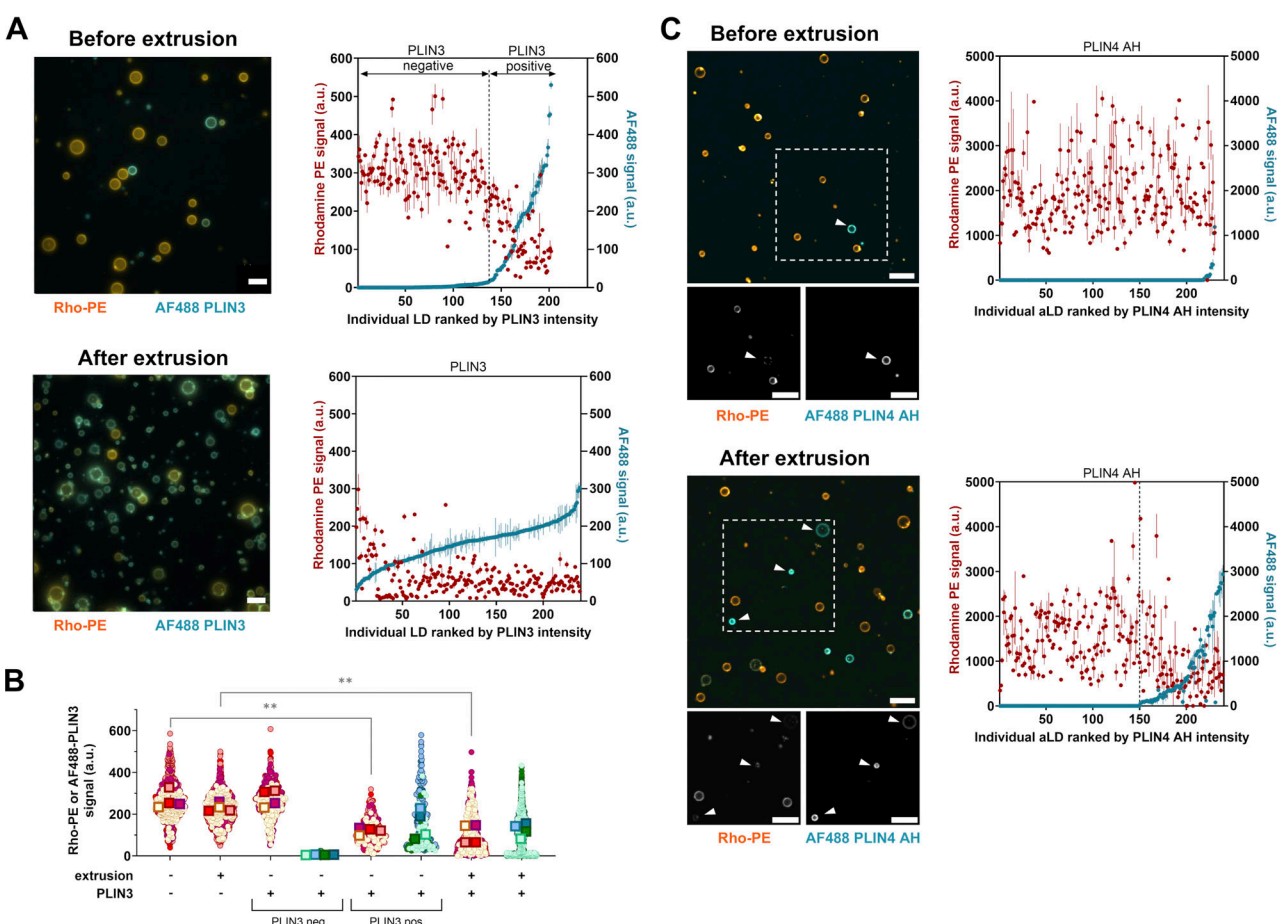

Figure 5. **Binding of PLIN3/4 to PC-covered aLDs negatively correlates with phospholipid coverage. (A)** Left: Typical wide field fluorescence microscopy images of PC(16:0/18:1)-covered aLDs (10 μm; with 0.5% Rho-PE shown in orange) in the presence of PLIN3 (with 10% AF488 PLIN3 shown in blue) before and after extrusion. Plots on the right show the individual fluorescence intensities of AF488 PLIN3 and Rho-PE on 100–250 aLDs as ranked according to AF488 PLIN3 intensity. **(B)** aLD surface analysis of the phospholipid and protein coverage from four independent experiments, one of which is shown in A. The condition "*before extrusion with PLIN3*" was separated in AF488-positive and AF488-negative aLDs, as shown in A. Data are shown as superplots (Lord et al., 2020). Each small circle corresponds to one LD. The large squares show the mean of each experiment. To compare Rho-PE changes, a two-factorial ANOVA was performed on all conditions, except the "*non-extruded with PLIN3-neg*"; PLIN3 factor $F_{(1,3)}$ = 73.9 and P value = 0.0033. For post-hoc comparisons of interest a Holm's test was performed, **P < 0.01. **(C)** Same analysis as in A with PLIN4 AH (12mer) except that the images were acquired with a confocal microscope. Scale bars: 10 μm.

Schneiter, 2023; Choi et al., 2023; Scherer et al., 1998). Therefore, we next asked whether binding of PLIN3 to lipid bilayers occurs by a phospholipid exclusion mechanism as observed with aLDs. A rare example of phospholipid exclusion in bilayers has recently been provided by the structure of caveolin, which occupies only one bilayer leaflet (Porta et al., 2022). We visualized PLIN3 on glass bead-supported lipid bilayers. In agreement with the NBD assays using liposomes (see Fig. 3 D and Choi et al., 2023), PLIN3 bound poorly to PC(14:0/14:0), PC(16:0/18:1), and PC(18:1/18:1)-supported bilayers, whereas strong binding was observed on supported bilayers made of PC(4Me) (Fig. 7 A). In contrast with our observations on aLDs, the Rho-PE signal and the AF488 PLIN3 signal coincided on all bead-supported bilayers, and we observed no significant decrease in the Rho-PE signal after PLIN3 binding to PC(4Me) bilayers (Fig. 7 A). The correlation between PLIN3 binding and low phospholipid coverage is therefore unique to the LD monolayer.

**Visualization of Arf1-GTP binding to LDs does not reveal phospholipid exclusion**

Next, we wanted to test whether other LD-binding proteins behaved similarly as PLIN3 in our in vitro binding assay. We chose the small G protein Arf1, which has been extensively studied at the molecular and cellular level (Donaldson and Jackson, 2011). When switching to the active GTP-bound state, Arf1 exposes a myristoylated AH for organelle interaction (Liu et al., 2010). In vitro, Arf1-GTP binds readily to many types of lipid bilayers as well as to PC-covered aLDs (Thiam et al., 2013a). In cells, activated Arf1 can be found on many organelles, including LDs; its exact subcellular localization depends on the localization of guanine nucleotide exchange factors (Bouvet et al., 2013; Wilfling et al., 2014). Moreover, binding of Arf1-GTP to LDs enables the recruitment of the COPI coat and the subsequent budding of nanodroplets. Such budding reduces the amounts of phospholipids at the LD surface, thereby inducing LD surface tension and facilitating

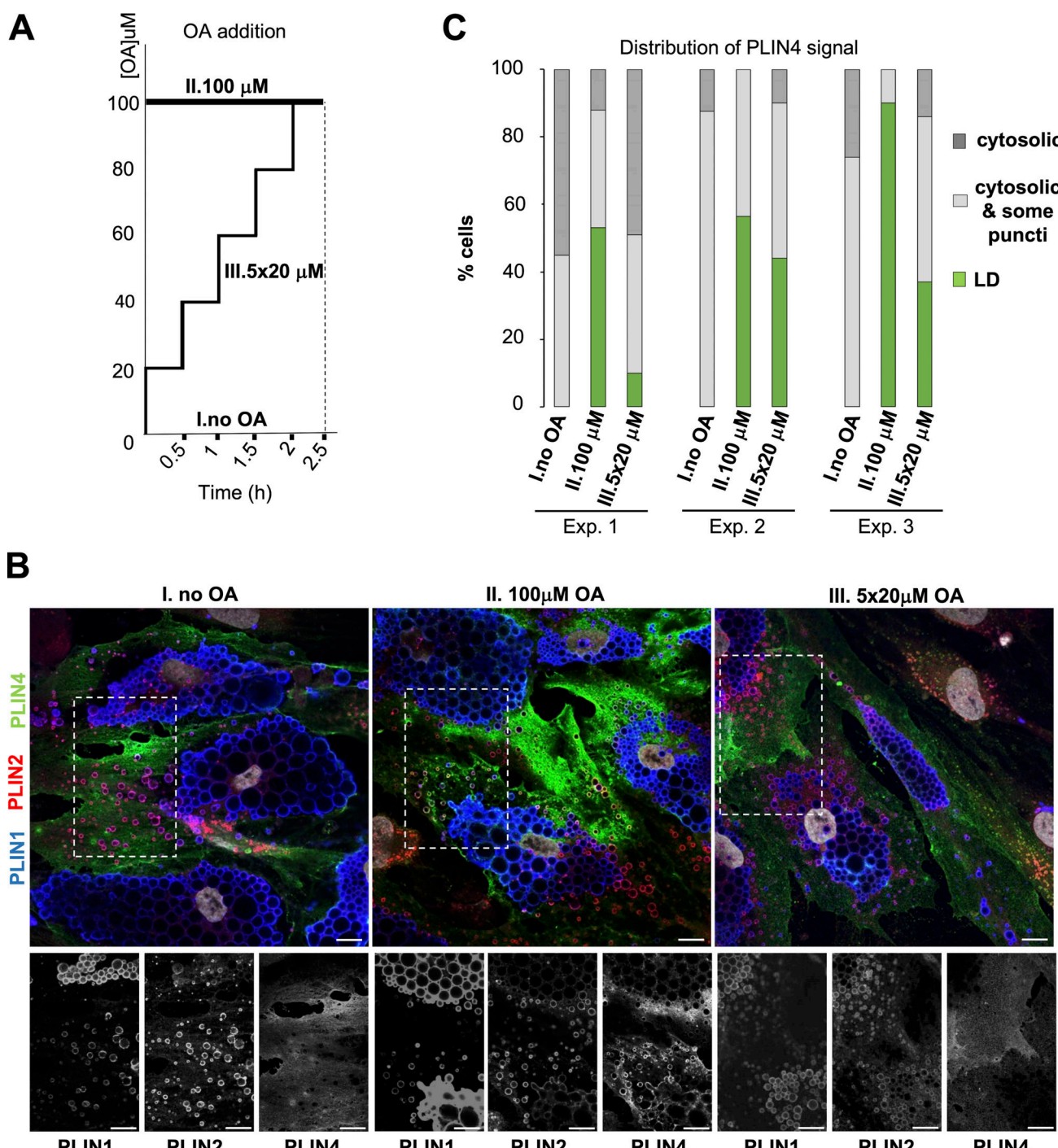

Figure 6. **Binding of endogenous PLIN4 to LDs in human adipocytes (TERT-hWA) depends on the rate of triglyceride synthesis. (A)** Protocol for the gradual or sudden increase of oleic acid in the culture medium of differentiated TERT-hWA adipocytes. **(B)** Immunofluorescence analysis showing representative z sections obtained by confocal fluorescence microscopy of endogenous PLIN1 (blue), PLIN2 (red), and PLIN4 (green) at 10 days of differentiation under the three conditions schematized in A. Scale bar: 10 μm. **(C)** Quantification of three independent experiments. The categories are the same as the ones defined in Fig. 1. Only fully differentiated cells containing large PLIN1-labeled LDs, which represent the majority of all cells at day 10 were included in the analysis. Number of cells counted in each experiment: Exp. 1: 23/61/53; Exp. 2: 66/46/67; Exp. 3: 37/43/49; numbers correspond to categories no OA/100 μM OA/5 × 20 μM OA, respectively.

the recruitment of other proteins to the LD surface (Thiam et al., 2013a; Wilfling et al., 2014). It was thus of interest to compare the aLD binding properties of Arf1-GTP and PLIN3 with respect to phospholipid coverage.

We used an Oregon-green C-terminally labeled form of Arf1 (Arf1-OG), which binds artificial liposomes in a GTP-dependent manner akin to authentic Arf1 (Manneville et al., 2008; Ambroggio et al., 2010). We incubated Arf1-OG with PC-covered

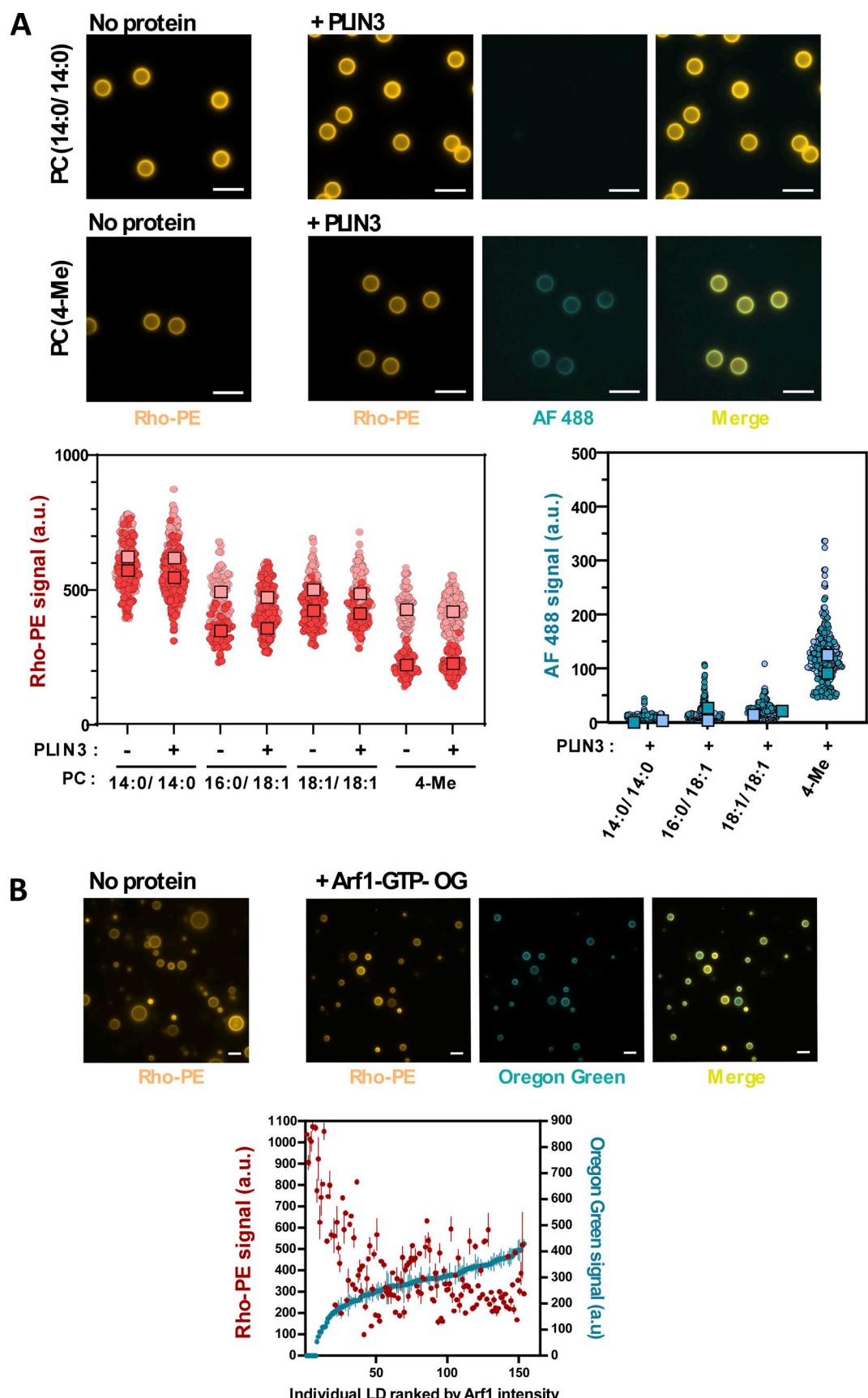

Figure 7. **Binding of PLIN3 to PC bilayers and of Arf1 to PC-covered aLDs does not correlate with change in phospholipid coverage. (A)** PLIN3 was incubated with 5 µm bead-supported bilayers made of PC of the indicated acyl chain composition. Images of Rho-PE and AF488 PLIN3 were taken using a

fluorescence microscope and quantified with a Fiji macro. Data are shown as superplots (Lord et al., 2020). Light and dark symbols distinguish two experiments. Each small circle is one bead. The large squares show the mean. **(B)** Arf1-OG was incubated for 1 h on POPC-covered aLDs (containing Rho-PE as a tracer) in the presence of an excess of GTPγS and in the presence of EDTA to promote nucleotide exchange. Thereafter images of Arf1-OG and Rho-PE were taken with a fluorescence microscope. The plot on the right shows the individual fluorescence intensities of Arf1-OG and Rho-PE on 150 aLDs ranked according to Arf1-OG intensity. Source data are available for this figure: SourceData F7.

aLDs and triggered the exchange between GDP and the non-hydrolyzable GTP analog GTPγS by lowering Mg²⁺ concentration. In contrast to our observation with PLIN3 before extrusion, most aLDs became intensely stained with Arf1-OG (Fig. 7 B). Variations in the Arf1-OG signal between aLDs was modest and there was no obvious negative correlation between the Rho-PE and the OG signals except for a minor fraction of aLDs that displayed a very high in Rho-PE signal and showed almost no Arf1 signal. We conclude that binding of Arf1 to aLDs is permissive to high PC monolayer coverage and that Arf1-GTP cannot distinguish between a monolayer and a bilayer.

Altogether, the experiments shown in Fig. 7 indicate that that the replacement of phospholipids by PLIN3 at the LD surface is unique to LDs and to specific proteins: neither the binding of PLIN3 to phospholipid bilayers nor the binding of Arf1-GTP to aLDs leads to phospholipid exclusion.

### LD tension promotes shallow lipid packing defects that are independent of PC acyl chain profile

In lipid bilayers, replacing saturated with monounsaturated phospholipids promotes the formation of packing defects, which become more abundant with increasing membrane curvature (Vanni et al., 2014). As visualized by molecular dynamics simulations, these defects appear deep, hence explaining the preferential adsorption of AH with large hydrophobic groups, such as ALPS motifs, to highly curved and monounsaturated membranes (Bigay et al., 2005; Vanni et al., 2014). Interestingly, previous simulations of ternary interfaces between TG(18:1/18:1/18:1), phospholipids and water revealed that surface tension promotes the formation of different lipid packing defects from those observed in bilayers (Bacle et al., 2017; Kim and Swanson, 2020; Prévost et al., 2018). These defects are wide but shallow and result from the interdigitation of TG(18:1/18:1/18:1) within the POPC monolayer. Such defects might be more adapted to PLIN AHs, which contain rather small hydrophobic residues (e.g., Ala, Val, and Thr) (Giménez-Andrés et al., 2021).

To extend this analysis, we performed MD simulations on ternary (TG(18:1/18:1/18:1)/PC/water) systems in which we varied both PC coverage (surface tension) and PC unsaturation (14:0/14:0, 16:0/18:1, and 18:1/18:1) (Fig. 8 A and Fig. S5). At high PC coverage (i.e., low tension), the surface of the PC monolayer was comparable with the surface of a bilayer, showing an increase in lipid packing defects according to the level of PC unsaturation (14:0/14:0 < 16:0/18:1 < 18:1/18:1). When we decreased PC coverage, the differences between the three monolayers decreased. Deep lipid packing defects increased and then plateaued, whereas shallow packing defects exponentially increased regardless of the actual unsaturation level of PC (Fig. 8 B). These defects corresponded to regions where TG(18:1/18:1/18:1) molecules were directly exposed to the solvent (Kim and Swanson, 2020) (Fig. 8 A and Fig. S5). Altogether, the simulations were in line with the experimental data and reinforce the conclusion that the acyl chain profile of phospholipids is less defining for the interfacial properties of LDs under tension as compared with lipid bilayers (Fig. 8 C). The LD surface packing properties depend mostly on the oil reservoir underneath, an effect that becomes prominent at low phospholipid coverage.

## Discussion

Although PLINs are the most abundant LD-associated proteins and were among the first to be identified, our understanding of their mode of interaction with LDs remains incomplete. Notably, the mechanisms enabling the various PLINs to localize to different LDs in the same cell are mysterious. Here, we succeeded in purifying full-length PLIN1/2/3 and developed a novel method of LD reconstitution, which allowed us to study PLIN-LD interactions under both equilibrium conditions and following mechanical perturbations of the LD surface. Chemical and physical changes of the LD surface are inherent features of LDs and may happen at different LD life stages, from lipogenesis to lipolysis (Olarte et al., 2021; Dhiman et al., 2020). We also improved LD analysis by mastering LD density and by using complementary methods for quantification. In particular, flow cytometry provides a high level of precision since it combines analysis at the single LD level with the handling of thousands of LDs in microliter samples.

Our major conclusion is that different PLINs show very different requirements for binding to LDs. On the one hand, the interaction of PLIN1 with LDs is very robust and is affected neither by changes in LD surface composition nor by mechanical perturbations of the lipid surface, allowing PLIN1 to stably associate with LDs during adipocyte differentiation. While this manuscript was in preparation, an article reported that PLIN1 contains two hydrophobic regions downstream of the AH that make the protein behave as an integral ER protein in the absence of LDs (Majchrzak et al., 2024). On the other hand, PLIN3 is very sensitive to large perturbations of the LD surface; its binding requires transient LD deformation (in our in vitro experiments promoted by LD extrusion), an effect that largely surpasses what can be achieved by changing the lipid composition of the LD surface, including the incorporation of DAG(18:1/18:1), which has a smaller headgroup than phospholipids. PLIN2 falls between these two extreme cases. The truncated version of PLIN4 used here suggests that, akin to PLIN3, PLIN4 is a low affinity LD protein, consistent with previous observations (Čopič et al., 2018; Giménez-Andrés et al., 2021). In addition, MD simulations show that packing defects in model LDs differ from those observed on bilayers, especially at high LD surface tension when the phospholipid coverage of LDs is only partial. Instead of

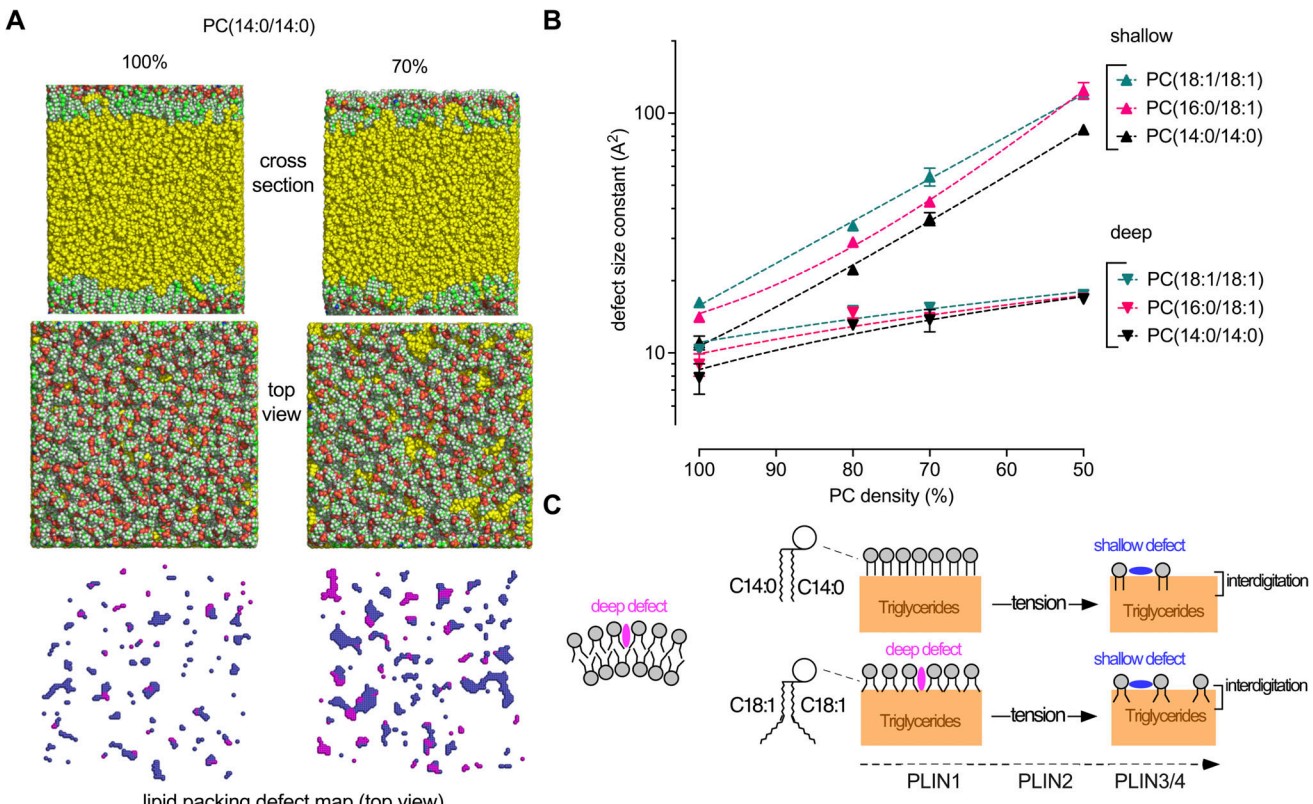

**Figure 8. Molecular dynamic simulations of ternary water/PC/triolein systems at increasing tension. (A)** Cross section, top view, and packing defect map of TG(18:1/18:1/18:1) (yellow) covered with a monolayer of PC(14:0/14:0) at 100% or 70% PC coverage (N: blue, C: green, O: red, H: white). The cartesian maps show the corresponding top view of deep (purple) and shallow (blue) defects (Gautier et al., 2018). For a complete view of the combined effects of tension and PC unsaturation, see Fig. S5. **(B)** Quantification of the packing defects as a function of PC coverage and unsaturation. Each point corresponds to one ternary water/PC/TG(18:1/18:1/18:1) system with the indicated % of PC. The exponential distribution of deep and shallow defects was converted into a characteristic area constant, expressed in Å² (Gautier et al., 2018). **(C)** Schematic view of the impact of surface tension and PC unsaturation on LD molecular surface. As surface tension increases, the underlying TG(18:1/18:1/18:1) molecules interdigitate with PC and become solvent exposed, thereby inducing the formation of shallow defects. These defects differentially condition binding of PLIN1–4 to LDs and are distinct from the deep lipid packing defects induced by curvature and mono-unsaturation on PC bilayers.

displaying deep cavities adapted to bulky hydrophobic residues, the LD surface at high tension shows shallow but wide lipid packing defects, which result from the interdigitation of the underneath oil molecules with the phospholipid monolayer (Prévost et al., 2018; Bacle et al., 2017; Kim and Swanson, 2020). This interdigitation reduces the impact of the phospholipid acyl chain profile on the LD surface properties and makes LDs under tension well adapted to host extended AH protein regions containing small hydrophobic residues, a hallmark of PLIN3 and PLIN4 (Čopič et al., 2018; Giménez-Andrés et al., 2021; Griseti et al., 2023; Dhiman et al., 2020). When we analyzed PLIN3 or PLIN4 AH on LDs, we observed a strong anticorrelation between protein and phospholipid densities, suggesting that these proteins essentially replace the phospholipid monolayer.

Interestingly, the two PLINs specific for mature adipocytes (Bäckdahl et al., 2021; Klingelhuber et al., 2024; Wolins et al., 2003; Barneda and Christian, 2017) are the most different: PLIN1 is not soluble whereas PLIN4 is a gigantic intrinsically disordered soluble protein (Griseti et al., 2023). In resting differentiated adipocytes, these contrasting properties translate into different localization. PLIN1 is present on the large pre-existing

LDs, whereas PLIN4 is in the cytosol. However, upon fatty acid addition, the new droplets become solely decorated by PLIN4. Fast growing LDs might be better handled by a low affinity soluble PLIN because it is immediately available. In addition, PLIN4 is adapted for coating LDs with its highly extended AH (Giménez-Andrés et al., 2021; Čopič et al., 2018). PLIN2 and PLIN3 are more ubiquitous. Given their difference in LD affinity, it is possible that they follow the same division of labor as PLIN1 and PLIN4, with PLIN2 coating mature droplets and PLIN3 handling fast forming ones.

Overall, the use of a repertoire of PLINs with different solubility and affinity for LDs might help cells to cope with LDs of different dynamics. For example, during the process of LD formation, it has been noticed that for an LD to quickly and faithfully bud toward the cytosolic side of the ER, there must be mechanisms to control its cytosolic surface. If not, the droplet might bud toward the luminal side (Chorlay et al., 2019; Choudhary et al., 2018; Nieto et al., 2023, *Preprint*). In cells, impeding or boosting phospholipid metabolism can indeed affect LD size and/or budding direction, which can be corrected by expressing PLINs (Čopič et al., 2018; Chorlay et al., 2019). In

simulations and biochemical reconstitutions, LD budding directionality can be controlled by adding new phospholipids or proteins to one bilayer leaflet in parallel with TG supply (Chorlay et al., 2019; Choudhary et al., 2018; Nieto et al., 2023, *Preprint*). Although in such in vitro systems, strong hydrophobic peptides appear more efficient, this is tempered by limitations in solubility (Chorlay et al., 2019). Our study suggests that the benefit of low affinity PLINs is to provide a highly soluble pool to specifically coat LDs under tension, e.g., during LD formation. These considerations might be also important for targeting of other proteins that distribute on a subset of LDs in the cell (Thul et al., 2017; Thiam and Beller, 2017); recent examples are spartin (Chung et al., 2023) and APOE (Windham et al., 2024).

The capacity to prepare artificial droplets covered with full-length perilipins should offer new possibilities to study their function, interactions, and differences. These include their ability to sense different oils, the dissection of the kinase-dependent lipolysis cascade on PLIN1 LDs, or the mechanism of CIDE-induced LD fusion. The importance of understanding the precise mechanisms of PLIN function on LDs is underscored by their diverse implications in different diseases. In particular, mutations in PLIN1 and PLIN4 have been linked to many metabolic phenotypes, and loss-of-function heterozygous mutations in PLIN1 and PLIN4 positively and negatively, respectively, correlate with metabolic disease (Duan and Savage, 2023; Griseti et al., 2023). Overexpression of PLIN2 and PLIN3 has been observed in a number of cancers and often correlates with higher cellular proliferation and poor prognosis (Safi et al., 2024). Overexpression of PLIN4 has been shown to promote drug resistance of triple-negative breast cancer (Sirois et al., 2019). These links are not surprising, given the central role of LDs in mediating cellular lipid metabolism and energy homeostasis. Mechanistic understanding of PLINs function on LDs has therefore important implications for human health.

## Materials and methods

### Adipocyte cell culture
Human adipocyte TERT-hWA cells (Markussen et al., 2017) were cultured using a protocol modified from Klingelhuber et al. (2024). Preadipocytes were cultured in dishes on glass slides in DMEM-F12 medium (12634010; Thermo Fisher Scientific) with 10% FBS (F7524; Sigma-Aldrich) and supplemented with 2.5 µg/ml ßFGF (F3685; Sigma-Aldrich). When cells reached confluence, serum was removed from the culture medium (day P). 2 days later (day 0 of differentiation) and until day 6, differentiation was initiated by the addition of an adipogenic cocktail containing 1 nM T3 (T6397; Sigma-Aldrich), 0.5 mM IBMX (I5879; Sigma-Aldrich), 5 µg/ml insulin (I6634; Sigma-Aldrich), 1 µM cortisol (H0369; Sigma-Aldrich), 1 µM dexamethasone (D4902; Sigma-Aldrich), and 1 µM rosiglitazone (R2408; Sigma-Aldrich). After day 6, the adipogenic cocktail was removed and fresh culture medium with 0% FBS was added to cells every 2 days.

At different points of the differentiation (days 3, 6, and 10), culture medium was supplemented with 100 µM oleic acid (O1383; Sigma-Aldrich) complexed with fatty acid-free BSA

(Sigma-Aldrich) for 2.5 h (in a single addition or in 20 µM increments added every 30 min), after which the cells were fixed and prepared for imaging.

### Preparation of protein extracts and western blot analysis
TERT-hWA cells cultured in 100-mm culture dishes were washed with PBS and harvested in 300 µl of ice-cold lysis buffer (150 mM NaCl, 50 mM Tris-HCl pH 7.4, 1% NP-40, 0.5% Na-deoxycholate, 0.1% SDS, and 2 mM EDTA containing "Complete mix" protease inhibitors from Roche) and passed 10 times through a 26-gauge needle. Total proteins were quantified using the Bradford assay and samples were denatured in Laemmli buffer at 95°C for 5 min. Proteins were separated on a 12% SDS-PAGE gel and transferred to a nitrocellulose membrane (GE Healthcare), which was then incubated overnight with primary antibodies followed by a 1-h incubation with fluorescent secondary antibodies (Table S1), which were detected using an Odyssey imaging system (Odyssey M; LI-COR).

### Immunofluorescence and confocal microscopy
Cells were fixed in 3.2% paraformaldehyde (28906; Thermo Fisher Scientific) for 15 min at room temperature (RT). They were then gently permeabilized with 0.5% saponin in phosphate-buffered saline (PBS) containing 0.5% FBS for 15 min at RT, washed with PBS, and incubated in a blocking solution containing PBS and 0.5% FBS for 30 min at RT. They were then incubated with primary antibody cocktails using antibody dilutions as indicated in Table S1, overnight at 4°C, rinsed in PBS, and incubated with secondary antibodies and DAPI stain (D1306; Thermo Fisher Scientific) for 1 h at RT, protected from light, and mounted for imaging.

Multidimensional images of fixed adipocyte cells were acquired at RT with a LSM980 confocal microscope (Zeiss), using a 40× Plan-Apo 1.4 NA oil-immersion objective. The microscope is equipped with T-PMT camera and driven by Zeiss Zen Blue software. Excitation sources used were 405 nm diode laser, an Argon laser for 488 and 514 nm, and a Helium/Neon laser for 633 nm.

### Synthetic genes and plasmid construction
Codon-optimized sequences of human PLIN1, PLIN2, and PLIN3 were ordered from Eurofins Genomics. They contained, upstream, a NdeI (5′-CATATG-3′) restriction site, a hexahistidine tag (5′-CATCATCACCATCACCAC-3′), a TEV site (5′-GAAAAC CTGTACTTCCAAAGC-3′) and, downstream, a Ssp1 restriction site. Genes were subcloned into a NdeI/Ssp1 digested pET16b.-His10.TEV.LIC expression vector (Jamecna et al., 2019). All constructs were controlled by DNA sequencing before transformation into *E. coli* strains.

### Protein expression and purification
The codon-optimized plasmid for hPLIN1 was expressed in *E. coli* ArcticExpress (DE3) (Agilent) while codon-optimized plasmids hPLIN2 and hPLIN3 were expressed in *E. coli* BL21-Gold (DE3). Cells were grown in LB medium with 50 µg/ml of ampicillin at 37°C in 2-L flasks to an $OD_{600\ nm}$ of ≈0.6 and induced with 1 mM isopropyl b-D-1-thiogalactopyranoside (IPTG) at 37°C for 3 h in

the case of hPLIN2 and hPLIN3. For hPLIN1, cells were cooled on ice for 30 min before induction with 1 mM isopropyl b-D-1-thiogalactopyranoside (IPTG) at 27°C for 1h 30 min. Cells were harvested by centrifugation for 20 min at 6,000 $g$ and stored at −20°C.

For the purification of PLIN1 and PLIN2, pellets were resuspended in buffer A (25 mM Tris, pH 7.5, 300 mM NaCl, and 30 mM imidazole) containing cOmplete EDTA-free protease inhibitor cocktail (Roche), 1.5 μM pestatin, and 2 μM bestatin. The cells were lysed using a cell disruptor, supplemented with 0.5 mM phenylmethylsulfonyl fluoride (PMSF), and centrifuged at 120,000 × $g$ at 4°C for 45 min. Pellets were resuspended in Buffer B (25 mM Tris-HCl, 300 mM NaCl, 30 mM Imidazole, 7 M Urea, pH 7.5) and centrifuged at 120,000 × $g$, at 4°C, for 30 min. The resulting supernatant was incubated at 4°C for 3 h with pre-equilibrated Co-NTA beads (Thermo Fisher Scientific). The unbound fraction was eluted from the beads and the beads were washed with 10 volumes of buffer B before elution with Buffer C (20 mM MES, 300 mM NaCl, 500 mM imidazole, 7 M Urea, pH 6.3). The eluted fractions were supplemented with 2 mM dithiothreitol (DTT), pooled, and concentrated on an Amicon 10 kDa MWCO cell. The protein pool was purified on a sephacryl S300 HR column (Cytiva) equilibrated with Buffer D (25 mM Tris, 120 mM NaCl, 3 M Urea and 2 mM DTT, pH 7.5). For 1 L of culture, the average yield of purification was 0.5 and 30 mg for hPLIN1 and hPLIN2, respectively.

For the purification of hPLIN3, the bacteria pellet was resuspended in buffer A supplemented with a tablet of cOmplete EDTA-free protease inhibitor cocktail (Roche), 1.5 μM pestatin, and 1.5 μM bestatin. Cells were lysed using a cell disruptor, supplemented with 0.5 mM of PMSF, and centrifuged at 120,000 $g$ at 4°C for 45 min. The resulting supernatant was incubated at 4°C for 3 h with pre-equilibrated Co-NTA beads (Thermo Fisher Scientific). The unbound fraction was separated from the beads, and the latter were washed with 10 volumes of buffer A before elution with Buffer E (25 mM Tris, 300 mM NaCl, and 500 mM imidazole, pH 7.5). The eluted fractions were supplemented with 2 mM of DTT, pooled, and incubated overnight with TEV (Tobacco Etch Virus) protease to remove the 6His-Tag. The pooled fraction was concentrated on an Amicon 10 kDa MWCO cell and applied to a sephacryl S300 HR column (GE Healthcare) equilibrated with Buffer F (25 mM Tris pH 7.5, 120 mM NaCl, and 2 mM DTT). The average yield of full-length hPLIN3 was 7 mg per liter of culture.

## Protein labeling

Labeling of PLIN3 was performed after purification. Following the gel filtration step, the protein was loaded on an Illustra NAP-5 column equilibrated with 25 mM Tris, pH 7.5, and 120 mM NaCl. Thereafter, PLIN3 was labeled on endogenous cysteines by incubation at room temperature with a 10-fold excess of maleimide AF488 C5 or IANBD amide (N,N0-dimethyl-N-(iodoacetyl)-N'-(7-nitrobenz-2-oxa-1,3-diazol-4-yl)ethylenediamine) for 15 min. The unconjugated probe was blocked with L-cysteine and removed by buffer exchange on a NAP-5 column.

In the case of PLIN1 and PLIN2, labeling was performed during purification. Proteins associated with Co-NTA beads

were incubated at room temperature with an excess of maleimide AF488 C5 for 15 min to label endogenous cysteines. The unconjugated probe was removed by washing the beads with buffer B before elution of the protein using buffer C. The labeled protein was further purified by size exclusion chromatography as described for the unlabeled form.

PLIN4 AH (12mer) was purified and labeled with AF488 C5 maleimide as described (Čopič et al., 2018). Myristoylated Arf1 C182-Oregon green was prepared as described (Manneville et al., 2008).

## Limited proteolysis

Limited proteolysis was carried out in 25 mM Tris pH7.4, 120 mM NaCl, 1 mM MgCl$_2$, and 1 mM DTT with a final concentration of 0 or 2 M Urea. Proteins (3 μM) were incubated at 25°C under agitation with 1 μg/ml of subtilisin. At indicated times, aliquots were withdrawn and the reaction was stopped by adding 2 mM PMSF.

## aLD preparation

All lipids were from Avanti polar lipids, except TG(18:1/18:1/18:1) and BVO (CAS: 8016-94-2), which were from Sigma-Aldrich and Spectrum Chemical MFG Corp, respectively. Contaminants such as free FA were removed from BVO following a method described elsewhere (Lebo et al., 2004). Briefly, polypropylene tubes were first washed with hexane for 24 h. Then, eight parts (ml) of methanol was added to 1 part of oil (g) and vortexed. The tubes were centrifuged at 1,650 $g$ for 5 min. They were then placed upright in the freezer (−20°C) for at least 8 h. The methanol was discarded and the tubes were left at RT to thaw the oil. New methanol was added and the process was repeated five times. Finally, the mix was dried in a rotavapor and stored at −20°C in an argon enriched atmosphere.

To prepare aLDs of defined density ($\rho$) and diameter ($d$), we relied on the following equations:

$$f_{BVO} = \frac{\rho - \rho_{TG}}{\rho_{BVO} - \rho_{TG}} = \frac{\rho - 0.91}{1.33 - 0.91} \qquad (1)$$

$$[PL] = 166 \frac{\%oil}{d} \qquad (2)$$

Eq. 1 gives the volume fraction of BVO as a function of TG density ($\rho_{TG}$), BVO density ($\rho_{BVO}$), and the actual oil density ($\rho$) of the mixture. For a density $\rho = 1.05$, which was used in most experiments, this gives a volume fraction of BVO and TG of 0.33 and 0.67, respectively.

Eq. 2 gives the phospholipid concentration ($[PL]$, in μM) that is needed to emulsify a defined percentage of oil in buffer ($\%_{oil}$) into aLDs with a defined average diameter ($d$, in μm). This equation derives from two calculations of the total surface of LDs ($S_{oil}$):

First, the total number of phospholipid molecules multiplied by their elementary surface ($A_{PL} \approx 0.7$ nm$^2$) gives the surface of phospholipid monolayer available:

$$S_{oil} = [PL]VN_{av}A_{PL} \qquad (3)$$

where $N_{av}$ is the Avogadro number ($6.02 \times 10^{23}$), $V$ is the volume of the emulsion, and $A_{PL}$ is the elementary surface of phospholipids.

Second, the total surface of aLDs is also the number of aLDs (*n*) multiplied by their elementary surface;

$$S_{oil} = n \pi d^2 \quad (4)$$

Because the number of aLDs, *n*, is the ratio between the oil volume and the elementary volume of the aLDs, this gives:

$$S_{oil} = \frac{\frac{\%oil}{100} V}{\frac{4}{3}\pi \left(\frac{d}{2}\right)^3} \pi d^2 = 6 \frac{\%oil}{100} \frac{V}{d} \quad (5)$$

Combining Eqs. 3 and 5 gives Eq. 2.

For a volume fraction of 0.75% oil in buffer, obtaining a suspension of aLDs with a calculated diameter of 2 µm requires a concentration of phospholipid [PL] = 62.5 µM. This concentration was used to prepare aLDs for flotation and flow cytometry measurements. For larger aLDs used in light microscopy experiments (calculated diameter 10 µm), we used a fivefold lower concentration: [PL] = 12.5 µM.

To prepare the aLDs, we first mixed 10 µl TG(18:1/18:1/18:1) (9.11 mg) and 5 µl BVO (6.6 mg) from stock solutions in chloroform (≈90 mg/ml). For aLDs used for flotation and flow cytometry experiments (2 µm aLDs), the mix was supplemented with 125 nmol of a chosen PC species (PC(14:0/14:0), PC(16:0/18:1), or PC(18:1/18:1)) and 0.625 nmol Rhodamine-PE, both as stock solutions in chloroform. For aLDs used for fluorescence microscopy experiments (10 µm LDs), the mix was supplemented with 25 nmol of a chosen PC species (PC(14:0/14:0), PC(16:0/18:1), or PC(18:1/18:1)) and 0.125 nmol Rhodamine-PE. After evaporation of chloroform under a stream of nitrogen, the final oil volume containing the phospholipids (10 µl TG and 5 µl BVO) was resuspended with 2 ml HKMD buffer (HEPES 50 mM pH 7.2, K acetate 120 mM, MgCl$_2$ 1 mM, and DTT 1 mM), hence leading to a 0.75% oil suspension. The suspension was vigorously vortexed for 2 min at maximum speed (small aLDs) or briefly vortexed and pipetted five times with a Hamilton syringe (large aLDs), and then extruded 19 times through 1-µm (for aLDs used in flotation and flow cytometry experiments) or 8-µm polycarbonate filters (for aLDs used in fluorescence microscopy experiments) using a hand mini extruder (Avanti). The aLDs were kept at room temperature under argon, protected from light, and used on the same day.

### Protein:lipid ratio in aLD binding measurements

The protein and the phospholipid concentrations were chosen to be compatible with the available aLD surface area. For the experiments conducted with small (2 µm) aLDs, the actual phospholipid concentration was 62.5 µM. An amphipathic helix of ≈100 aa has a length of 0.15 × 100 = 15 nm and a width of about 1 nm, hence a surface of ≈15 nm$^2$. The average surface of a phospholipid is in the range of 0.7 nm$^2$. Therefore, a 100-aa helix occupies a surface comparable with 20 phospholipids. To have the helix and the protein on equal footing in term of potential LD coverage would give 62.5/20 ≈ 3 µM protein. In effect, we used 1–2 µM (hence a ≈1:50 mol:mol ratio) to also take into account the fact other regions besides the central 11-mer repeats of perilipins might contribute to binding or at least occupy a surface above the aLD surface. The same reasoning applies to the aLDs used for light microscopy observations: here, the

phospholipid concentration was 12.5 µM and the protein concentration was 0.25 µM, hence a 1:50 mol:mol ratio.

### aLD flotation on sucrose gradients

500 µl of 1 µm–extruded aLDs (0.75% oil) covered with the indicated PC (14:0/14:0, 16:0/18:1, or 18:1/18:1; concentration = 62.5 µM and 0.5 mol% Rho-PE) were supplemented with protein (e.g., PLIN3) among which 5–20% was fluorescently labeled in a total volume of 600 µl. Half of the sample was then further extruded 19 times through a 1-µm pore size polycarbonate filter. 150 µl of the non-extruded or extruded samples were then mixed with 100 µl sucrose (75% wt/vol) in HKMD buffer in a centrifuge tube. The resulting 30% sucrose cushion was overlaid with 250 µl sucrose (25% wt/vol in HKMD buffer) and 50 µl HKMD buffer. After centrifugation for 1 h at 55,000 rpm and at 20°C in a TLS 55 (Beckman) rotor, three fractions (top, middle, bottom) were collected using a Hamilton syringe and analyzed by SDS-PAGE using direct fluorescence or Sypro-staining and by Rhodamine fluorescence to determine protein binding to aLDs and aLD recovery.

### Flow cytometry experiments

Similar samples as those used in flotation and microscopy experiments were prepared and analyzed by flow cytometry. The samples containing 8 or 1 µm aLDs were diluted 10 or 50 times, respectively, in HKMD buffer to a final volume of 500 µl. Samples were acquired with a Cytek Aurora (Cytek Biosciences) equipped with five lasers (355, 405, 488, 561, 640 nm) and 64 detectors. 50 µl of each sample was recorded at medium flow rate to assess aLDs concentration and determine fluorescence intensity and percentage of aLDs with bound protein. Only particles within the typical S shape area of the scattering diagrams ISS(IFS) were analyzed (see Fig. S2 C). Data were unmixed in SpectroFlo v3.0.1 (Cytek Biosciences) and analyzed in FlowJo v10.9 (BD Biosciences).

Similar flow cytometry experiments were performed using an Attune CytPix flow cytometer, equipped with a brightfield camera, to measure aLD size. After gating the areas of interest, a total of 10,000 images were taken per sample and first processed in the Attune Cytometric software v6.0.1. Images with aggregated aLDs were eliminated and the remaining images were analyzed in batches (per condition per gated region) using a dedicated pipeline for Cell Profiler v4.2.1 and v4.2.5. The minimum Feret diameter was chosen to avoid manual screening and elimination of images containing aggregated aLDs that might have passed through the first selection filter.

### FRET experiments to study the impact of extrusion on aLD coverage by phospholipids

Experiments were performed in a JASCO fluorimeter at 25°C using a rectangle 3 × 10-mm quartz cuvette. aLDs (2 µm) were prepared by extrusion through a 1-µm polycarbonate filter. They contained 100% triolein and were covered with PC(16:0/18:1) and with 0.25–1.5 mol% of NBD-PE and Rho-PE. Emission spectra (465–750 nm; scan speed 500 nm/min; excitation of NBD at 455 nm) were recorded before and after applying a second extrusion through a 1-µm polycarbonate filter. Excitation and

emission bandwidths were 5 nm. For kinetic measurements, the experimental conditions were similar except that the measurements were performed at constant excitation and emission wavelengths (455 and 590 nm, respectively) with a sample rate of one measurement/second.

## Dynamic light scattering (DLS) analysis
The same protein/aLD mixtures as those used for flotation or flow cytometry experiments were analyzed by DLS without dilution using a VASCO KIN particle size analyzer (Cordouan technologies).

## Lipid extraction and HPTLC
Lipids were extracted from 100 µl of small aLDs, before and after extrusion with protein. Because in aLDs the ratio PL/TG is very low, a three-phase liquid extraction (3PLE) was performed to separate polar and neutral lipids in different organic phases. The method and solvent ratios used were adapted from Vale et al. (2019), Shibusawa et al. (2006). Briefly, the separate solvents were added to the sample and then the aqueous phase (sample) was completed with water, resulting in Hex:EtAc:ACN:Aqueous (3:1:3:2). On the day of extraction, 50 µg/ml BHT was added to each solvent separately. Samples were vortexed and centrifuged at 2,500 g for 4 min at 20°C. A fixed volume of upper phase was collected and hexane was added (half the volume of the first extraction) to the two remaining phases for re-extraction. Samples were again vortexed and centrifuged, and fixed volumes of upper and middle phases were collected separately. To reduce PL loss, a re-extraction of middle phase was done with ACN and EtAc (3:1) (half the volume of first extraction). Samples were again vortexed, centrifuged, and collected.

Hex (upper) phases were dried under nitrogen. ACN (middle) phases were dried in a vacuum centrifuge (speedvac).

An automatic TLC Sampler 4 (ATS4; CAMAG) with a spray needle was used to apply 6 and 15 µl of Hex and ACN samples, respectively, and 3 µg of each standard (manually mixed, Avanti Polar Lipids and Sigma-Aldrich; BVO from Spectrum Chemical) onto a Merck HPTLC glass plate silica gel 60 (20 × 10 cm, layer thickness 200 µm). The syringe was washed three times between samples in CHCl$_3$:MeOH (50:50).

The plate was then eluted with eight different solvent mixes. The first five elutions were performed in an automated multiple development chamber (AMD2; CAMAG), with preconditioning in MeOH. The following three elutions were done in an automatic developing chamber (ADC2; CAMAG) with a fixed humidity percentage achieved with a saturated MgCl$_2$*5H$_2$O solution. Saturation without pads was done for 10 min before the two first elutions, and plate preconditioning was done for 5 min. The plate was dried for 5 min between elutions. Solvent systems' % distribution by order of elution: Ethyl acetate, 1-propanol, chloroform, methanol, 0.25% (wt/vol) aqueous potassium chloride, (1) 24:30:27:11:8, (2) 27:27:27:11:8, (3) 27:27:27:19:0, all up to 50 mm; (4) Ethyl acetate, chloroform (50:50), up to 55 mm; (5) Ethyl acetate, chloroform (30:70), up to 60 mm; (6) Hexane, ethyl acetate (60:40), up to 70 mm; (7) toluene up to 78 mm; and (8) hexane up to 85 mm.

Plate surface was sprayed with a modified copper sulphate solution (Handloser et al., 2008) in a derivatization chamber (CAMAG) and revealed by heating in a plate heater (CAMAG) under a fume hood. Imaging was done in a Fusion FX7 instrument (Vilber Loumat) using epi white light for the charred lipids and fluorescence detection for Rho-PE (before charring). Lipid identification was carried out by comparison to standards applied onto the same TLC plate.

Plate images were analyzed in Fiji by determining peak areas corresponding to each lipid band of interest.

## NBD fluorescence measurements with liposomes
The experiments were performed essentially as in Čopič et al. (2018). Dry films containing chosen lipids were prepared from stock solutions in chloroform. The film was resuspended in 50 mM HEPES, pH 7.2, and 120 mM K-acetate at a concentration of 5 mM lipids. After five cycles of freezing in liquid nitrogen and thawing in a water bath, the multilamellar liposomes were extruded through 100-nm pore size polycarbonate filters. Fluorescence emission spectra upon excitation at 505 nm were recorded in a Jasco RF-8300 apparatus. The sample (600 µl) was prepared in a cylindrical quartz cell containing liposomes (0, 50, 150, or 450 µM lipids) in HK buffer supplemented with 1 mM MgCl$_2$ and 1 mM DTT. The solution was stirred with a magnetic bar and the temperature of the cell holder was set at 37°C. After acquiring a blank spectrum, NBD-PLIN3 (150 nM) was added and a second spectrum was measured and corrected for the blank. The fluorescence ratio of NBD-PLIN3 at 540 nm in the presence of liposomes versus in solution was then determined.

## Protein binding to bead-supported lipid bilayers
To prepare bead-supported bilayers, extruded liposomes (100 nm, 250 µM lipids) were incubated with 5 × 10$^6$ uniform silica beads of 5 µm in diameter (Bangs Laboratories, Inc.) in HKMD buffer (100 µl total volume) for 30 min at room temperature under gentle mixing. The beads were washed three times in HKMD buffer and centrifuged (200 g for 2 min) before observation. Liposomes were prepared as described in the previous section, with four different PC compositions, all presenting 0.5% mol of Rho-PE. 5 µl of the total bead-supported bilayers prepared were used. 100% of AF488-PLIN3 was added to the bilayers in Ibidi chambers at 37°C to obtain a final concentration of 3.2 nM in a total volume of 300 µl. Final protein concentration was calculated by taking into account the area of exposed PL layer on a bead and the total number of beads used. Images were taken with an EMCCD Camera (iXon Ultra 897; Oxford Instruments) on an inverted wide-field fluorescent microscope (IX83; Olympus) using a 60×/1.42 oil immersion objective and operated with MetaMorph (Molecular Devices).

## Fluorescence microscopy of aLDs
450 µl of 8 µm-extruded aLDs (0.75% oil) covered with the indicated PC (14:0/14:0, 16:0/18:1, or 18:1/18:1; concentration = 12.5 µM) were supplemented with 250 nM PLIN3 (10% was labeled with AF488) or PLIN4 AH (12 mer, of which 10% was fluorescently labeled). The total volume was 500 µl. Half of the sample was then further extruded 19 times through a 8-µm

polycarbonate filter. aLDs were diluted ≈5 times in HKMD buffer in Ibidi slides at 37°C. Slides were previously passivated with free-FA BSA (6 mg/ml) and washed three times with buffer before use. For PLIN3, images were taken with an EMCCD Camera (iXon Ultra 897, Oxford Instruments) on an inverted wide-field fluorescent microscope (IX83; Olympus) using a 60×/1.42 oil immersion objective and operated with MetaMorph (Molecular Devices). Observation of aLDs with PLIN4 AH was performed with an inverted Olympus Ixplore Spin SR microscope coupled with a spinning disk CSU-W1 head (Yokogawa) using 60X UPLXAPO 1.42 NA DT 0.15 mm oil-immersion objective. Z stacks of 10 planes with a step size of 0.5 µm were acquired with an ORCA Fusion BT Digital CMOS camera (Hamamatsu) using 488 and 561 nm 100 mW lasers and GFP narrow (520/10) and mCherry (593/40) filters. The system was driven by CellSens Dimension 3.2 software.

For experiments with myristoylated Arf1-OG, 50–100 nM protein was incubated in a total volume of 200 µl in HKMD buffer in the presence of 20–40 µl of 8 µm-extruded aLDs (0.75% oil) and with an excess of GTPγS (40 µM). Nucleotide exchange at the surface of the aLDs was promoted by the addition 2 mM EDTA, and droplets were observed in Ibidi slides as for PLIN3.

### Image analysis

For aLDs, the images of Rho-PE and AF488 channels were analyzed using a custom-made macro in Fiji. A line was manually drawn across each droplet on composite images. Both fluorescent profiles per aLD, per field, were obtained automatically. The background was subtracted and two maximal intensity values, per aLD, of the two channels were registered. The mean of these two values, corresponding to the two intersections between the line and the LD contour, was used for graphical representation (superplots) and further analyses. For very small aLDs, only one maximum value was obtained. The analysis was performed in a minimum of three different fields per condition.

Selected single z-section images of adipocytes cells were analyzed manually using Image J.

### Molecular dynamic simulations

All-atom simulations were performed using the forced field Charmm36. The triolein (TG(18:1/18:1/18:1)) topology was modified as in Campomanes et al. (2021). We started from bilayers (14.1 × 14.1 nm) containing 400, 412, or 440 molecules of PC(18:1/18:1), PC(16:0/18:1), or PC(14:0/14:0) in water (50 Å). We incorporated 864 molecules of triolein between the two monolayers and then performed minimization and equilibration for 110 ns using GROMACS v2023.4. Simulations were conducted for 600 ns. Thereafter, we determined the size distribution of lipid packing defects using PackMem (Gautier et al., 2018).

### Statistical analysis

Statistical tests were performed with GraphPad Prism v10 and InVivoStat v4.10.

### Online supplemental material

Fig. S1 shows immunofluorescence characterization of TERT-hWA cells. Fig. S2 shows the impact of extrusion on the properties of aLDs in term of PLIN3 binding, aLD composition, and light scattering properties. Fig. S3 shows the purification and intrinsic properties of PLIN1, PLIN2, and PLIN3. Fig. S4 shows the size analysis of aLDs upon PLIN3 addition and/or extrusion. Fig. S5 shows the surface properties of aLDs as a function of PC coverage as inferred from molecular dynamics. Table S1 shows antibodies used in this study.

### Data availability

Data are available in the primary article and the supplementary materials. Raw data files for all flow cytometry experiments are available upon request from the authors.

## Acknowledgments

We thank Jacob B. Hansen (University of Copenhagen, Copenhagen, Denmark) for the gift of the TERT-hWA cells, Jacques Fattaccioli for input on flow cytometry experiments, Sophie Abelanet for help with image analysis, Thomas Lorivel for advice on statistical analysis, Bayane Sabbagh for PLIN4 12mer purification, and Niklas Mejhert, Dominique Langin, Scott Frendo-Cumbo, and Cyril Moulin for comments on the manuscript. We acknowledge SABLESPlatformes financed by the European Union through the European Regional Development Fund for the Aurora cytometer and the flow cytometry and microscopy facility from the Institut de Pharmacologie Moléculaire et Cellulaire, which is part of the «Microscopie Imagerie Cytométrie Azur» GIS IBiSA labeled platform. We also acknowledge the imaging facility MRI (Montpellier, France), member of the national infrastructure France-BioImaging supported by the French National Research Agency (ANR-10-INBS-04, «Investments for the future»).

This study was funded by the European Research Council (ERC Synergy 856404, SPHERES) and by the Agence National de la Recherche within the project entitled Investissements d'Avenir UCAJEDI (ANR-15-IDEX-01).

Author contributions: A.R. Dias Araujo: Formal analysis, Investigation, Methodology, Validation, Visualization, Writing—review & editing, A.A. Bello: Formal analysis, Investigation, Methodology, J. Bigay: Conceptualization, Formal analysis, Investigation, Methodology, Resources, Validation, Visualization, Writing—review & editing, C. Franckhauser: Investigation, Validation, Visualization, R. Gautier: Methodology, Software, J. Cazareth: Formal analysis, Methodology, D. Kovács: Conceptualization, Data curation, Formal analysis, Investigation, F. Brau: Formal analysis, Resources, N.P. Jean-Paul Fuggetta: Investigation, Visualization, A. Copic: Conceptualization, Data curation, Funding acquisition, Investigation, Methodology, Project administration, Supervision, Validation, Writing—review & editing, B. Antonny: Conceptualization, Funding acquisition, Methodology, Project administration, Supervision, Validation, Writing—original draft, Writing—review & editing.

Disclosures: The authors declare no competing interests exist.

Submitted: 12 March 2024

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

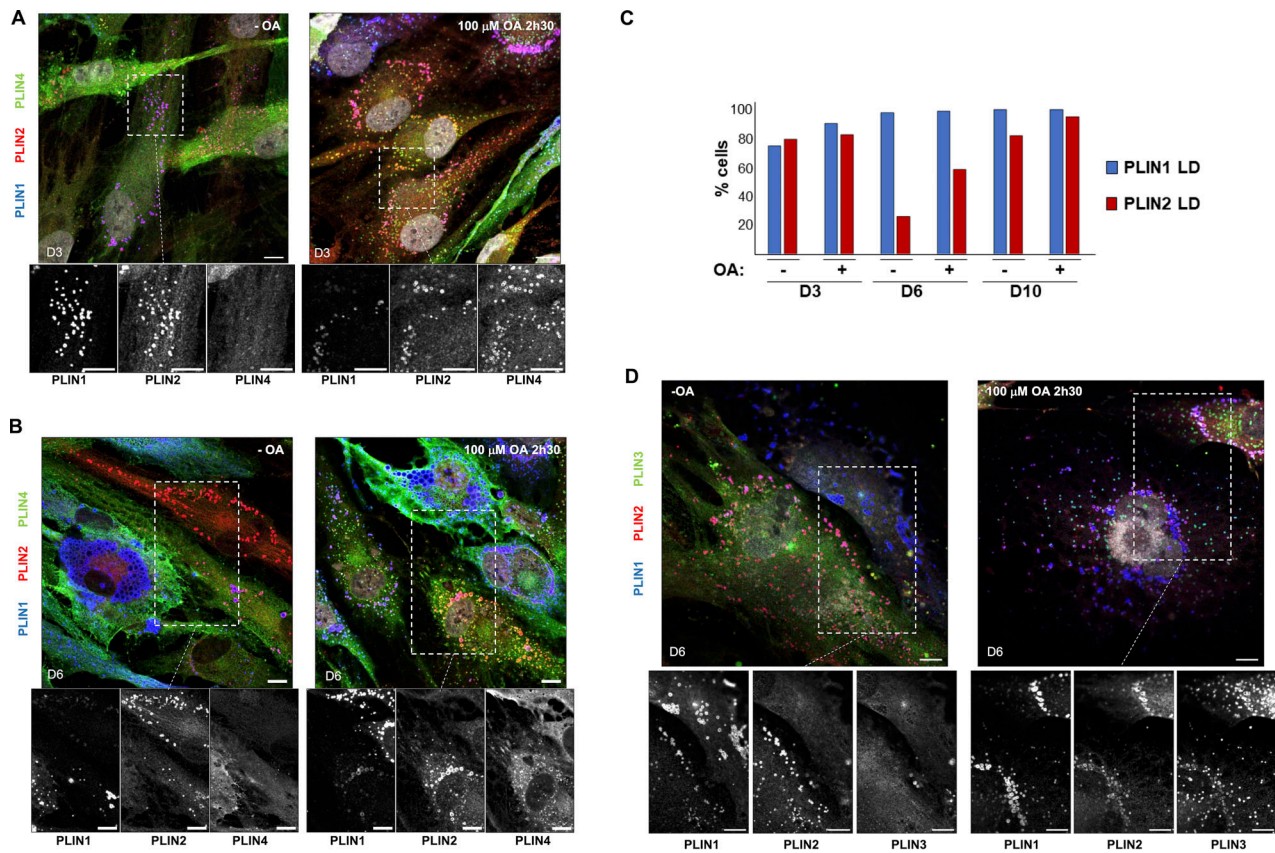

Figure S1. **Immunofluorescence analysis of perilipins in human adipocytes during differentiation and effect of oleic acid treatment. (A and B)** Representative z sections obtained by confocal fluorescence microscopy of endogenous PLIN1, PLIN2, and PLIN4 in human adipocytes at day 3 (A) and 6 (B) of differentiation after immunofluorescence with specific antibodies. Cells were either left in culture medium or fed with 100 µM oleic acid for 2.5 h before observation. Scale bar: 10 µm. **(C)** Quantification of the fraction of cells in which the PLIN1 and PLIN2 signals localized to LDs at D3, D6, and D10 of differentiation. In the rest of cells, the signal was either not detected (PLIN1) or appeared cytosolic (PLIN2). N of cells quantified was the same as in Fig. 1 D, from one of two representative experiments. **(D)** Same as in A and B (day 6 of differentiation in human adipocytes) for endogenous PLIN1, PLIN2, and PLIN3 in human adipocytes at day 6 of differentiation. Scale bar: 10 µm.

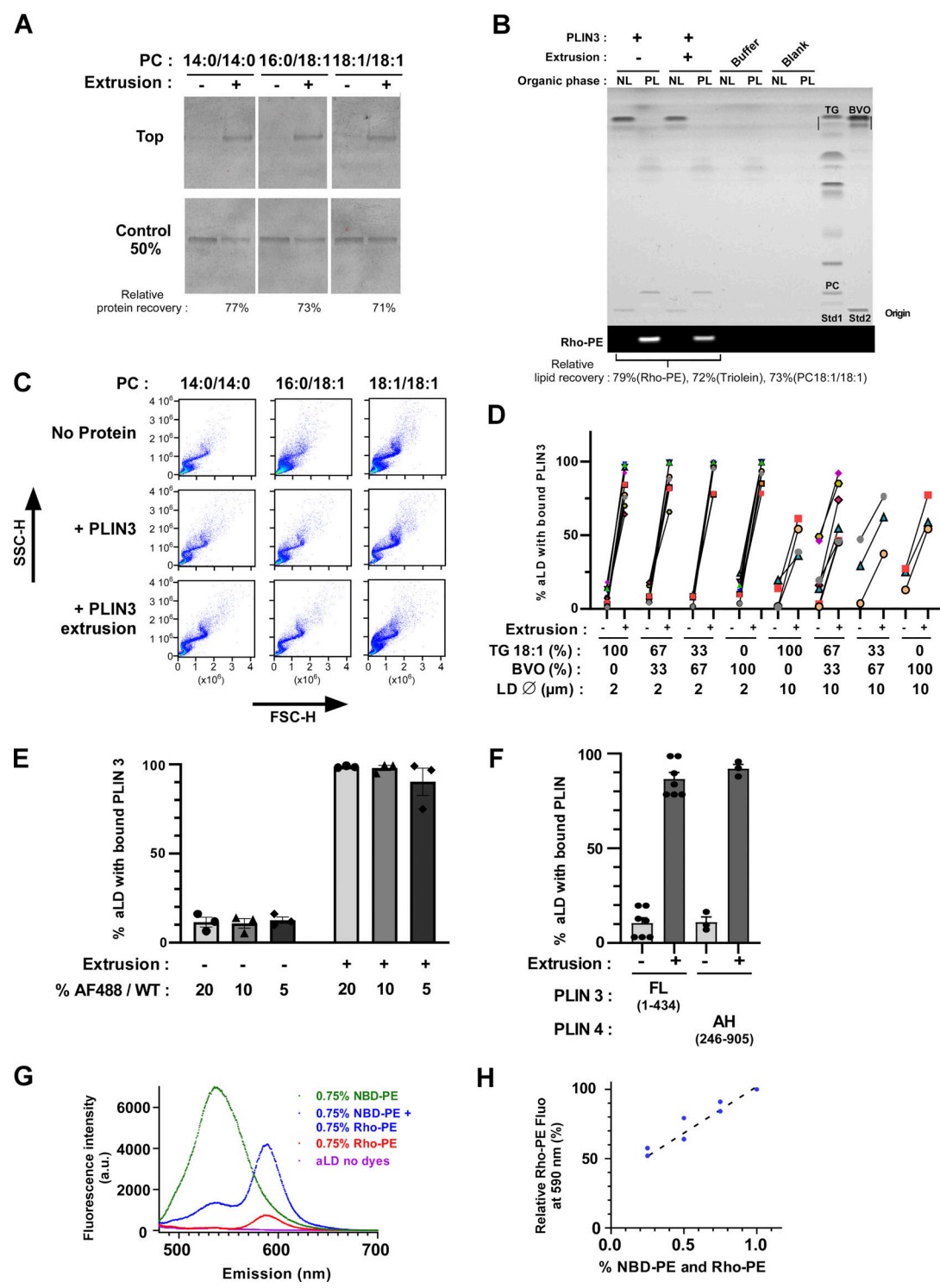

Figure S2.  **Binding of PLIN3 and the amphipathic region of PLIN4 to aLDs requires surface tension. (A)** Effect of extrusion on the binding of PLIN3 to PC-covered lipid droplets was assessed by separating aLDs from soluble protein by flotation followed by Sypro-orange staining to detect unlabeled PLIN3 in the aLD-containing top fraction. Extrusion: 1 μm. **(B)** HPTLC analysis of the PLIN3 + aLD sample, before and after extrusion (1 μm). NL, neutral fraction. PL, polar fraction. **(C)** Side scattering (SSC-H) versus forward scattering (FSC-H) diagrams of the same aLDs as that shown in Fig. 2 C. **(D)** Flow cytometry analysis of PLIN3 binding to PC-covered aLDs prepared with varying ratios of triolein and BVO. Droplet size: 2 μm or 10 μm. P values were calculated using a paired *t* test. n.s., P > 0.05, *P < 0.05, **P < 0.01, ***P < 0.001, ****P < 0.0001. **(E)** Flow cytometry analysis of PC-covered aLDs in the presence of varying ratios of AF488 PLIN3 versus unlabeled PLIN3. **(F)** Flow cytometry analysis of the binding of PLIN3 and of the AH region of PLIN4 on PC-covered aLDs. **(G)** Fluorescence emission spectra of PC(16:0/18:1)-covered aLDs (2 μm) containing no fluorescent lipids, 0.75 mol% NBD-PE, 0.75 mol% Rho-PE, or 0.75 mol% NBD-PE and Rho-PE. **(H)** Fluorescence at 590 nm of PC(16:0/18:1)-covered aLDs containing increasing % of NBD-PE and Rho-PE. The measurements were performed with decreasing amounts of aLDs to compensate for the increase in % of NBD-PE and Rho-PE, thereby keeping the concentration of the two fluorescent lipids constant. Source data are available for this figure: SourceData FS2.

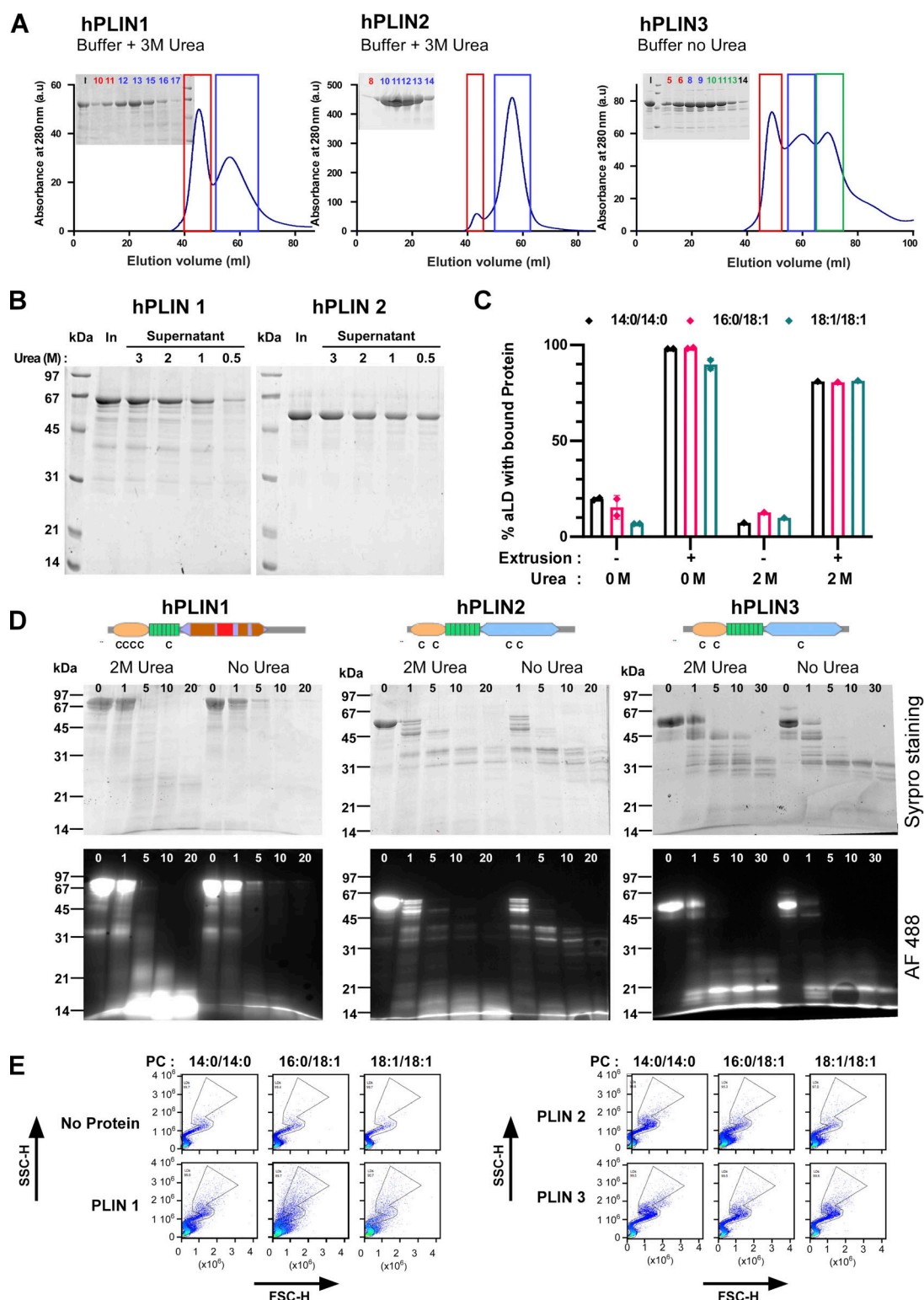

Figure S3.   **Purification and biochemical characterization of full-length human PLIN1, PLIN2, and PLIN3. (A)** Last step of purification of PLIN1-3 on a Sephacryl S300 HR column. In the case of PLIN1 and PLIN2, the buffer contained 3 M urea. **(B)** Sedimentation assay to determine the concentration of urea (in M) at which PLIN1 and PLIN2 remain soluble. In: input. **(C)** Flow cytometry analysis of PLIN3 + PC-covered aLDs in the presence or absence of 2 M urea. The experimental conditions were as in Fig. 2, B and C. **(D)** Time course of limited proteolysis of AF488 PLIN1, AF488 PLIN2, and AF488 PLIN3 at 0 or 2 M urea in the presence of subtilisin (time in min). The same gel as in Fig. 4 A is shown here but visualized both by direct fluorescence and after Sypro orange staining. The upper drawings show the predicted domain organization of PLIN1/2/3 and the localization of endogenous cysteines. **(E)** Side scattering (SSC-H) versus forward scattering (FSC-H) diagrams of the same aLDs as shown in Fig. 3 A in the absence or in the presence of the indicated PLINs. With PLIN1 and PLIN2, the scattering diagrams lost part of the typical S shape observed with oil emulsions made of individual droplets (Fattaccioli et al., 2009), suggesting partial aggregation of the particles. Source data are available for this figure: SourceData FS3.

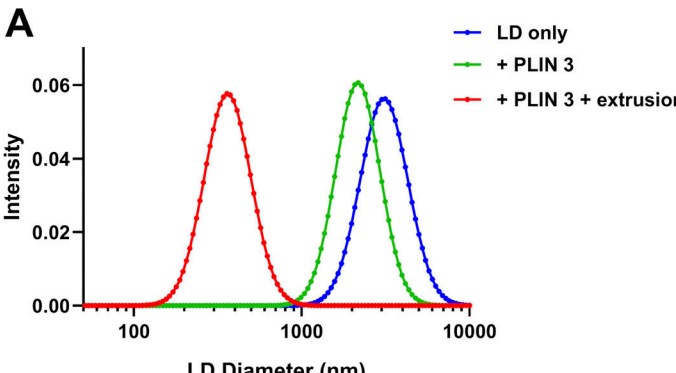

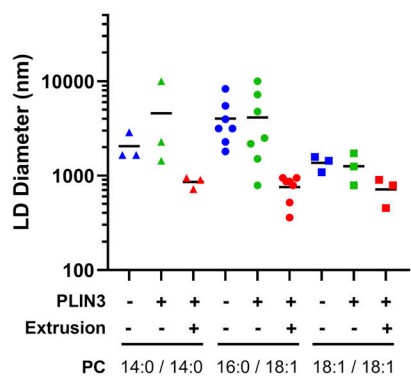

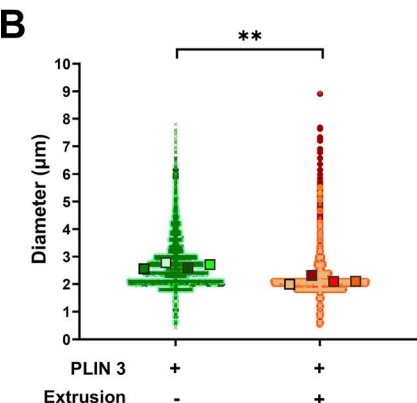

Figure S4. **aLD size analysis. (A)** DLS analysis of PC-covered aLDs (calculated diameter 2 µm) in the absence or in the presence of PLIN3 and before and after extrusion (1 µm). The experimental conditions were similar to those used in Fig. 2, B and C. The left panel shows the typical size distribution in one experiment using PC(16:0/18:1)-covered aLDs. The right panel shows the mean diameter as determined from three to seven independent experiments similar to that shown in A. **(B)** Size analysis using a Cytpix flow cytometer equipped with a brightfield camera to measure aLD size. PC(16:0/18:1)-covered aLDs (calculated diameter 10 µm) were incubated with PLIN3 and eventually extruded (8 µm). After gating, the AF488 negative region was compared with the AF488 positive region, before and after extrusion, respectively, which correspond to the regions where most aLDs are found (see e.g., Fig. 2 C). Data are shown as superplots (Lord et al., 2020). Each small circle is one droplet. The large squares show the means from four independent experiments. **P < 0.01 determined by a paired t test.

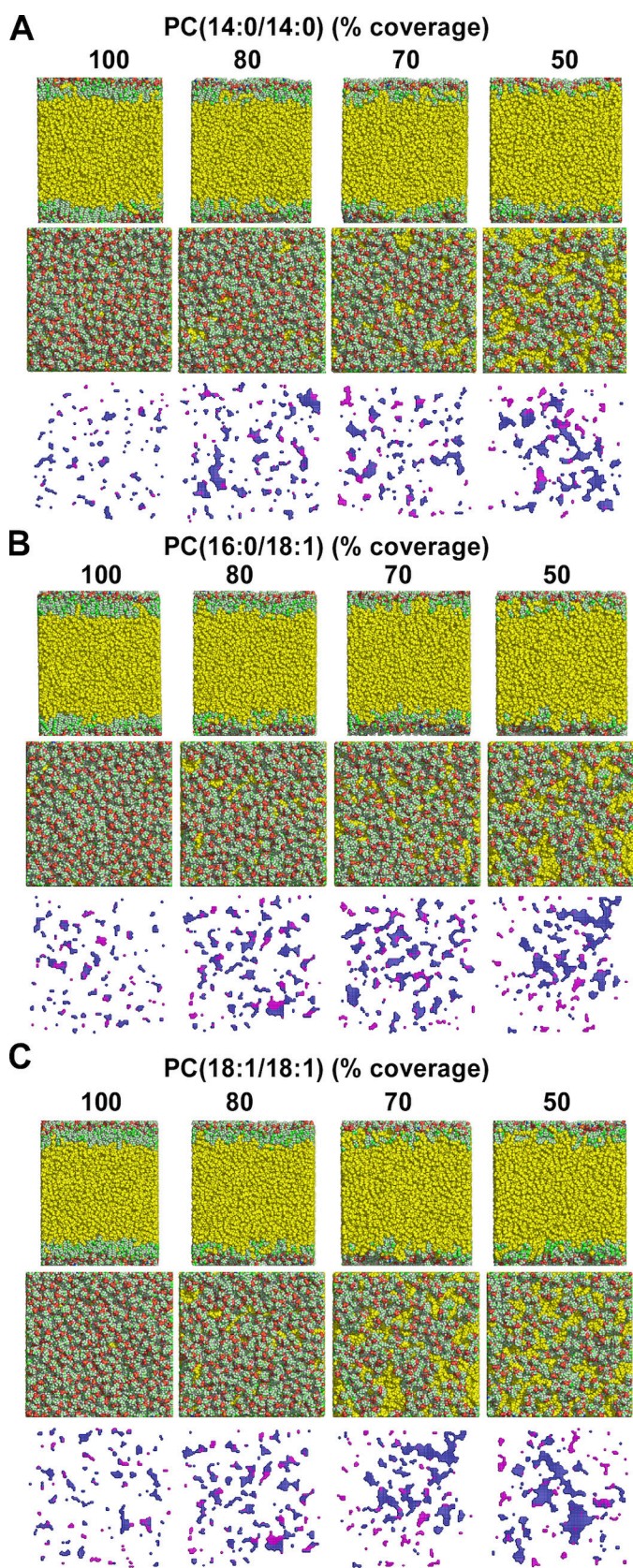

Figure S5.   **Molecular dynamic simulations of ternary water/PC/triolein systems at increasing tension. (A–C)** Top and side views of TG(18:1/18:1/18:1) covered with a monolayer of PC(14:0/14:0) (A), PC(16:0/18:1) (B), or PC(18:1/18:1) (C) at decreasing PC coverage. The cartesian maps show the corresponding top view of deep (purple) and shallow (blue) defects. Note the interdigitation between triolein (yellow) and phospholipids at low PC coverage. The difference in lipid packing defects between saturated and monounsaturated monolayers vanishes as PC coverage decreases.

**Provided online is Table S1. Table S1 shows antibodies used in this study.**

