## [Peer Review File · The Journal of Cell Biology]

Surface tension-driven sorting of human perilipins on lipid droplets

Ana Rita Dias Araujo, Abdou Akim Bello, Joëlle Bigay, Céline Franckhauser, Romain Gautier, Julie Cazareth, Dávid Kovács, Frédéric Brau, Nicolas Fuggetta, Alenka Copic, and Bruno Antony

Corresponding Author(s): Bruno Antony, Institut de Pharmacologie Moléculaire et Cellulaire; Abdou Akim Bello, Institut de Pharmacologie Moléculaire et Cellulaire; and Joëlle Bigay, Institut de Pharmacologie Moléculaire et Cellulaire

Review Timeline:

Submission Date:	2024-03-12
Editorial Decision:	2024-04-19
Revision Received:	2024-08-13
Editorial Decision:	2024-08-14
Revision Received:	2024-08-20

Monitoring Editor: William Prinz

Scientific Editor: Andrea Marat

Transaction Report:

DOI: <https://doi.org/10.1083/jcb.202403064>

April 19, 2024

Re: JCB manuscript #202403064

Dr. Bruno Antony
Institut de Pharmacologie Moléculaire et Cellulaire
Institut de Pharmacologie Moléculaire et Cellulaire
Université Côte d'Azur, CNRS, Inserm, UMR7275
660 route des Lucioles
Valbonne 06560
France

Dear Dr. Antony,

Thank you for submitting your manuscript entitled "Surface tension-driven sorting of human perilipins on lipid droplets". The manuscript was assessed by expert reviewers, whose comments are appended to this letter. We invite you to submit a revision if you can address the reviewers' key concerns, as outlined here.

All three reviewers found your study interesting and well done. They have thoughtful, constructive suggestions for how it might be improved. We invite you to consider addressing all their concerns, though we understand several of the suggested experiments may not be feasible. The most important suggestions to consider experimentally addressing are: (1) testing the prediction that the rate of LD growth affects PLIN targeting in cells (Rev 3, pt1), (2) quantifying packing defects on LDs (Rev 2, pt 1), (3) testing the effects of more physiologically relevant phospholipid compositions (Rev 3, pt 2) and excess phospholipids (Rev 1, major point) on PLIN targeting, and 4) determining whether brominated oil in LDs affects the results (Rev 2, pt 4).

GENERAL GUIDELINES:

Text limits: Character count for a Article is < 20,000, not including spaces. Count includes title page, abstract, introduction, the joint Results & Discussion, and acknowledgments. Count does not include materials and methods, figure legends, references, tables, or supplemental legends.

Figures: Articles may have up to 5 main text figures. To avoid delays in production, figures must be prepared according to the policies outlined in our Instructions to Authors, under Data Presentation, <https://jcb.rupress.org/site/misc/ifora.xhtml>. All figures in accepted manuscripts will be screened prior to publication.

*****IMPORTANT:** It is JCB policy that if requested, original data images must be made available. Failure to provide original images upon request will result in unavoidable delays in publication. Please ensure that you have access to all original microscopy and blot data images before submitting your revision. ***

Supplemental information: There are strict limits on the allowable amount of supplemental data. Articles may have up to 3 supplemental figures. Up to 10 supplemental videos or flash animations are allowed. A summary of all supplemental material should appear at the end of the Materials and methods section.

Please note that JCB now requires authors to submit Source Data used to generate figures containing gels and Western blots with all revised manuscripts. This Source Data consists of fully uncropped and unprocessed images for each gel/blot displayed in the main and supplemental figures. Since your paper includes cropped gel and/or blot images, please be sure to provide one Source Data file for each figure that contains gels and/or blots along with your revised manuscript files. File names for Source Data figures should be alphanumeric without any spaces or special characters (i.e., SourceDataF#, where F# refers to the associated main figure number or SourceDataFS# for those associated with Supplementary figures). The lanes of the gels/blots should be labeled as they are in the associated figure, the place where cropping was applied should be marked (with a box), and molecular weight/size standards should be labeled wherever possible.

The typical timeframe for revisions is three to four months. While most universities and institutes have reopened labs and

allowed researchers to begin working at nearly pre-pandemic levels, we at JCB realize that the lingering effects of the COVID-19 pandemic may still be impacting some aspects of your work, including the acquisition of equipment and reagents. Therefore, if you anticipate any difficulties in meeting this aforementioned revision time limit, please contact us and we can work with you to find an appropriate time frame for resubmission. Please note that papers are generally considered through only one revision cycle, so any revised manuscript will likely be either accepted or rejected.

Thank you for this interesting contribution to Journal of Cell Biology. You can contact us at the journal office with any questions at cellbio@rockefeller.edu.

Sincerely,

William Prinz, PhD
Monitoring Editor

Andrea L. Marat, PhD
Senior Scientific Editor

Journal of Cell Biology

Reviewer #1 (Comments to the Authors (Required)):

In this elegant study, the authors establish a new in vitro system to study the association of perilipins (PLINs) to lipid droplets (LDs) to study the binding characteristics of distinct perilipins. They demonstrate that PLIN1, PLIN2, and PLIN4 have different binding characteristics and selectivity for lipid droplets in cells, and establish a new assays to score perilipin 1-4 binding via FACS before and after application of physical stresses to induce membrane tension, which is highly relevant to LD biology. This study elegantly establishes remarkable differences in PLIN binding to lipid droplets. PLIN3/4 shows little tendency to bind to LD surfaces unless membrane tension is induced (mechanically or metabolically), while PLIN1, for example, constitutively binds to small lipid droplets and independent of membrane tension.

This study is highly original and contains new, elegant, original biochemical approaches. Overall, the study is very well written, and the technical quality of the presented data is very high. The technical advance presented in this paper will be of high relevance for everyone working in the field of lipid droplets and beyond. With only minor suggests for changes/additional explanations, I recommend this manuscript for publication in JCB.

Major point: Given that the elegant, new LD binding assay lies at the heart of this study: Is it possible to artificially generate LD with an excess of phospholipids? If possible, does this overabundance counteract tension by providing a reservoir of lipids, which should limit the binding of PLIN3 induced by extrusion. If this optional experiment is technically feasible (of which I am not entirely sure), it would be an important control for the revised manuscript.

Minor point, related to the previous one: Extrusion seems like a rather harsh way to induced tension (as well discussed by the authors). Is it possible to mimic metabolic activity in vitro by adding e.g ethanol (or other compounds) to the LD suspension? Alternatively, does prolonged vortexing (no extrusion) suffice to promote PLIN3 binding to LDs? It would be helpful if less harsh possibilities for inducing membrane tension and/or mimicking metabolic activity would be explored. Nevertheless, I would find such experiments very helpful, but not essential.

Other points:

P12-13 (and at other section of the manuscript): The authors frequently refer to phospholipid density. A consistent use of 'LD coverage with phospholipids' or 'phospholipid coverage' (as used by the authors) would be more suitable, clearly distinguishing it from lipid packing/lipid packing density at the molecular scale?

Related to the abstract: The following section is not entirely clear and should be rephrased. 'Binding of purified PLIN3 and PLIN4 AH was dependent on tension, even with polar lipids favoring packing defects, and showed an inverse correlation between protein and phospholipid densities on LDs. In contrast, PLIN1 bound readily to LDs fully covered by phospholipids; PLIN2 showed an intermediate behavior.' Without further context and before reading the manuscript, it is not becoming sufficiently clear what is meant by 'even with polar lipids favoring packing defects' (why 'even'). Also, it is unclear which inverse correlation is meant. What does 'fully covered by phospholipids' mean in this context, and what is meant by intermediated behavior.

Related to Figure 1C: Despite the text 'In striking contrast, PLIN4 displayed diffuse cytosolic signal in almost all non-treated cells at all stages of differentiation, with only faint staining of some peripheral LDs in a fraction of cells. Upon addition of OA, PLIN4 relocalized to peripheral LDs in a large fraction of cells (50% or more), with only a low number of cells retaining only cytosolic signal (Figure 1C).' I cannot find information on PLIN4 in Figure 1C. Have these data been represented in Figure 1D? Headlines to the individual panels might guide the eye or the reader and supporting an immediate, intuitive understanding of the figure.

Related to Figure 1/S1: It would help to move figure S1A forward in Figure 1. This would make Figure 1 immediately more 'accessible'.

Figure S1: The number of days of differentiation should be given in the Figure itself to improve the intuitive readability of the figure.

p9 L22: 'High performance thin layer chromatography (HPTLC) (Fig. S2B) and SDS-PAGE (Figure 2B) showed that the majority ($\approx 75\%$) of lipids and of protein was recovered after extrusion.' It is not entirely clear if this statement holds for all components equally: Proteins, phospholipids, and TAG. What is the % recovery of fluorescently labeled PE? Could those be provided with the Figure?

Related to P11 I262-263: The interesting observation that DOG does not promote PLIN3 binding in this experimental setup should be discussed (here or in the discussion) in light of earlier observations by Skinner et al., 2009; Ben M'barek et al., 2017; Choi et al., 2023; Stribny and Schneider, 2023, which are cited in the original manuscript. Can the seemingly divergent observations be reconciled?

Overall, a very interesting study overcoming a series of technical challenges! Very suitable for the readership of JCB.

Reviewer #2 (Comments to the Authors (Required)):

The study conducted by Araujo et al. aims to delve into the mechanisms underlying the specific targeting of Plin 1 to 4 to distinct subsets of LDs, particularly in human adipocytes. Specifically, they seek to unravel how Plin1 predominantly localizes to large and mature LDs, typically located in the cell center, while Plin2-4 target smaller LDs primarily situated at the periphery, especially upon oleate loading. After revisiting a long-standing observation, Figure 1, the authors introduce a novel assay integrating FACS, flotation, and imaging techniques, as depicted in Figure 2. This innovative method involves droplet extrusion to enhance the surface-to-volume ratio, thereby increasing monolayer packing defects, and subsequently monitoring the recruitment of purified Plin1-4. The study explores the influence of DPPC, POPC, and DOPC on protein recruitment, as well as the impact of LD size. While numerous research groups, including the authors' group, have already devoted tremendous efforts to elucidate the factors governing PLIN protein gradual binding, the techniques developed in this study are notably compelling, meticulously executed, and promising for a better understanding of PLIN membrane association. However, this reviewer feels a lack of clarity in connecting the findings across the figures. The authors suggest that differential LD tensions are responsible for the binding hierarchy of Plin1 >> Plin2 >> Plin3. This referee fails to see data supporting this notion. Nonetheless, this reviewer supports the publication of the study but strongly recommends that the authors consider addressing the following concerns, by order of priority.

1- The authors posit that LDs in differentiated hWA exhibit varying degrees of tension, which likely relates to lipid packing defects, with smaller LDs displaying a higher depletion of phospholipids compared to larger ones. This tension disparity is purported to influence the differential binding of Plin1-4 to LDs of varying sizes. Given this premise, it would be pertinent for the authors to quantify the level of packing defects in these LDs. One potential approach could involve utilizing packing defect sensors within cells; with Bruno Antony's group being the pioneer in developing and mastering such sensors, this could/should be doable. These sensors could potentially detect differences in phospholipid coverage between LDs, shedding light on the underlying mechanisms driving Plin1-4 binding. Alternatively, the authors could opt to purify LDs, followed by trypsinization or not, before exposing them to Plins. For instance, the binding of Plin3 to specific small purified LDs, if observed, would provide compelling evidence for the existence of tension discrepancies between LDs and further validate the authors' hypothesis.

2- While the authors tested the POPC and DOG mixture in 2D, it is noted that LDs are not exclusively composed of POPC. Therefore, conducting experiments with adequate phospholipid mixtures that better reflect LD composition, along with DOG, would provide more comprehensive information on the impact of DOG. This assay is pivotal, as the authors conclude that PLIN3 does not bind spontaneously to LDs even when surrounded by a DOG-enriched lipid monolayer, while it binds avidly to LDs under tension. However, this conclusion seems to contradict previous studies that have shown the localization of Plin3 to nascent LDs containing DOG (Khaddaj et al (PMID: 37465010), Choi et al (PMID: 37268630) and Chung et al (PMID: 31708432)).

3- This referee is unable to understand the "replacement terminology" model by the authors and the Arfgap data appears irrelevant to the manuscript. An alternative understanding of the data is that Plin3 exhibits better binding to LDs with lower

rhodamine signal due to the extrusion process leading to daughter LDs with varying densities of phospholipids. Therefore, Plin3 may have a preference for binding to LDs with reduced phospholipid coverage, rather than displacing phospholipids. The experiments with beads may not be directly comparable, as the lipid density is affected by methylation, allowing Plin3 to bind to beads with more space, such as the methylated phospholipid beads. Further clarification and detail are needed to support the notion that Plin3 displaces phospholipids, as proposed in the experiments and rationale provided.

4- The use of brominated vegetable oil at a concentration of 33% represents a substantial proportion. Given that bromine and hydrogen have markedly different solubility in water a priori, there is a potential discrepancy in the molecular architecture of the brominated oil at the LD interface compared to the triacylglycerols utilized in this study. Although the authors acknowledged this issue in figure S2, the impact of bromine before extrusion appears evident, particularly in the 10 μ m-sized LDs as observed figure S2D. To address this point, the referee encouraged the authors to conduct simple experiments comparing the use of 100% brominated oil versus 100% triacylglycerols in aLDs. Demonstrating that the presence of bromine does not significantly influence the results would provide crucial validation for the experimental findings.

5- A key aspect is missing from the manuscript, which is the range of tension alteration induced by the extrusion process. It may be extremely useful to quantify the extent of tension change when one droplet is divided into two through extrusion for example. Providing quantitative values for tension alteration would enhance understanding the experimental setup and its implications. Additionally, the authors could compare their values with those recently measured by the Ditscher group (PMID: 37212777) to provide further context and validation. Including such quantitative data would strengthen the rigor and interpretation of the findings.

Minor.

- Line 404 "At high PC coverage density (i.e. low tension), the surface of the PC monolayer was comparable to the surface of a bilayer, showing an increase in lipid packing defects according to the level of PC unsaturation (14:0/14:0 < 16:0/18:1 < 18:1/18:1)."

The authors need to provide clearer clarification regarding the contribution of their simulation study in comparison to the existing work by Farese/Walther, Vanni, and Swanson groups. Specifically, they should delineate how their simulation adds value or insight beyond what has already been explored by these groups. This clarification will help readers understand the unique perspective or findings offered by the authors' simulation with the existing body of literature.

- Line 414 "The LD surface packing properties depend mostly on the oil reservoir underneath, an effect that becomes prominent at low phospholipid density."

This is a sufficient rationale for conducting experiments specifically testing the impact of brominated lipids at 100% concentration.

- Please clarify whether the protein concentration was in excess relative to the available surface area of the aLDs

141 "a fraction of cells. Upon addition of OA, PLIN4 relocalized to peripheral LDs in a large fraction 142 of cells (50% or more), with only a low number of cells retaining only cytosolic signal (Figure 1C)." Should be 1D.

Figure 1D needs improvement as it is currently difficult to comprehend.

Reviewer #3 (Comments to the Authors (Required)):

This manuscript by Araujo et al investigates the targeting of perilipin proteins to lipid droplets (LDs), predominantly using a novel in vitro approach. Major conclusions are that PLIN1 behaves very different from other perilipins, and that extrusion as a model for a temporary increase in surface tension due to deformation of in vitro formed LDs allows some perilipins to bind LDs that otherwise would not bind LDs.

Overall, the manuscript reports well done and well controlled experiments, is well written (with the exception of referencing) and reports results for an interesting area of cell biology.

Weaknesses in my opinion are that:

- i) extrusion is a hard to control process with somewhat unclear consequences.
- ii) the main result, PLIN1 behaves different than the other PLINs, appears a bit trivial since the protein does not only have amphipathic helices used by other PLINs to bind LDs but also large, membrane embedded stretches of amino acid sequence
- iii) a key aspect of emerging model (PLIN binding is responsive to the speed of LD expansion) is not tested in cells.

Overall, this is an interesting and well performed manuscript with some limitations that could be better discussed. The main

points, I would recommend to address are:

- 1) Can the authors test whether varying the speed of LD formation (e.g., varying oleate concentrations and measuring LD growth and TG synthesis, using inhibitors of TG synthesis as controls added at different time points) affects PLIN targeting in cells? To me this would be important (and possible quite simple?) to add.
- 2) While the conclusions on the extrusion experiments are very reasonable, I wonder whether an orthogonal method (e.g. with a drop tensiometer) could be used to control surface tension and assay binding?
- 3) References in the manuscript seem a bit random and often appear to not well consider some of the older, original literature. The authors might want to carefully check this.
- 4) A procedural, but important point: all reported results in the paper need quantitation (absent of most of the cell biology experiments) and appropriate statistics (which seems not reported for any of the results); for an example lacking either, see figure 1. This is essential in my opinion.

Editor

All three reviewers found your study interesting and well done. They have thoughtful, constructive suggestions for how it might be improved. We invite you to consider addressing all their concerns, though we understand several of the suggested experiments may not be feasible. The most important suggestions to consider experimentally addressing are:

(1) testing the prediction that the rate of LD growth affects PLIN targeting in cells (Rev 3, pt1)

To address this point, we conducted new experiments using TERT-hWA cells. We compared the partitioning of perilipins under two conditions: when we added oleic acid as a single burst (100 μ M) or when we added the same amount of fatty acid in a stepwise manner (5 x 20 μ M). The results, which are shown in the **new Figure 6**, indicates that PLIN4 preferentially binds to LDs under the first condition. This experiment provides cellular evidence for our model; namely that PLIN3/4 are adapted to binding LDs under tension conditions.

New Figure 6

(2) quantifying packing defects on LDs (Rev 2, pt 1)

We present a new approach based on FRET to quantify the effect of extrusion on the phospholipid coverage of artificial droplets (aLDs) (new **Figure 2D-F**). We incorporated a few mol% of NBD-PE and Rho-PE in the PC(16:0/18:1) monolayer. Before extrusion, we observed a large FRET between the two probes (**Figure 2D-F** and **FigS2G**). Immediately after extrusion, we observed a large decrease (- 30 %) in the FRET signal, which recovers within 30 seconds (half-time \approx 15 s). Calibrating the FRET signal (new supplementary **FigS2H**), suggests that this drop corresponds to a two-fold decrease in the proximity between the two probes. When, we performed the same experiment in the presence of PLIN3, the recovery of the FRET signal was much lower. These

experiments indicate that extrusion facilitates PLIN3 binding by causing the transient spreading of the phospholipid monolayer.

New panels in Figure 2

New panels in Fig. S2

(3) testing the effects of more physiologically relevant phospholipid compositions (Rev 3, pt 2) and excess phospholipids (Rev 1, major point) on PLIN targeting, We performed new experiments in which we used aLDs covered by a phospholipid mixture similar to what has been reported for cellular lipid droplets by lipidomic in two papers: PC 68%, PE 25% and PI 7% (PMID: 22872753, PMID: 17210984). We also tested the impact of diacylglycerol in this context. Spontaneous binding of PLIN3 to these complex LDs was very low and similar to that observed with PC-covered LDs. Importantly, PLIN3 binding dramatically increased after extrusion for all lipid compositions. These experiments strengthen our conclusion that LD tension, not composition, is the major parameter governing PLIN3 binding. The corresponding experiments are shown in the new **Figure 3D**. We also tested the effect of excess phospholipids (see response to Rev1).

4) determining whether brominated oil in LDs affects the results (Rev 2, pt 4). We performed new experiments in which we gradually increased the fraction of brominated oil (0, 33, 66 and 100%) at the expense of regular oil (triolein). This substitution did not affect PLIN3 binding, which remained highly dependent on the extrusion step (**new Fig S2D**).

Please note that we have submitted this manuscript as an article. We have followed the guideline for this format: “Articles should be no more than 40,000 characters (not including spaces, figure legends, methods, or references), with up to 10 figures and/or tables. Articles may have up to five supplemental figures and references are unlimited”. The manuscript has 39262 characters, 8 main figures and 5 supplementary figures.

Reviewer #1:

In this elegant study, the authors establish a new in vitro system to study the association of perilipins (PLINs) to lipid droplets (LDs) to study the binding characteristics of distinct perilipins. They demonstrate that PLIN1, PLIN2, and PLIN4 have different binding characteristics and selectivity for lipid droplets in cells, and establish a new assays to score perilipin 1-4 binding via FACS before and after application of physical stresses to induce membrane tension, which is highly relevant to LD biology. This study elegantly establishes remarkable differences in PLIN binding to lipid droplets. PLIN3/4 shows little tendency to bind to LD surfaces unless membrane tension is induced (mechanically or metabolically), while PLIN1, for example, constitutively binds to small lipid droplets and independent of membrane tension. This study is highly original and contains new, elegant, original biochemical approaches. Overall, the study is very well written, and the technical quality of the presented data is very high. The technical advance presented in this paper will be of high relevance for everyone working in the field of lipid droplets and beyond. With only minor suggests for changes/additional explanations, I recommend this manuscript for publication in JCB.

Thank you for these very positive comments.

Major point: Given that the elegant, new LD binding assay lies at the heart of this study: Is it possible to artificially generate LD with an excess of phospholipids? If possible, does this overabundance counteract tension by providing a reservoir of lipids, which should limit the binding of PLIN3 induced by extrusion. If this optional experiment is technically feasible (of which I am not entirely sure), it would be an important control for the revised manuscript.

We tried the proposed experiment by including small PC liposomes in the LD extrusion step. This addition did not interfere with PLIN3 binding to LDs, which remained dramatically increased by extrusion as seen in the following flow cytometry data.

However, please note that the spontaneous fusion of liposomes to the LDs that are experiencing transient tension through extrusion might be not fast enough. Indeed, we now report new experiments by FRET that suggest that the LD suspension relaxes back to normal (probably by LD fusion and other shape rearrangements) within 30 seconds after extrusion (**New Figure 2D-F**, a copy is shown in our response to point #2 of the editor).

Minor point, related to the previous one: Extrusion seems like a rather harsh way to induced tension (as well discussed by the authors). Is it possible to mimic metabolic activity in vitro by adding e.g ethanol (or other compounds) to the LD suspension? Alternatively, does prolonged vortexing (no extrusion) suffice to promote PLIN3 binding to LDs? It would be helpful if less harsh possibilities for inducing membrane tension and/or mimicking metabolic activity would be explored. Nevertheless, I would find such experiments very helpful, but not essential.

Vortex has been successfully used before to promote the binding of PLIN4 fragments to pure oil (Copic et al 2018 Nat Commun). However, vortex and sonication depend strongly on the bench apparatus. Extrusion by the use of standard hand-extruders and defined pore-size polycarbonate filters offers a more reproducible way to mechanically stress LDs. In the revised manuscript, we better quantify the effect of extrusion on LD surface properties. Using FRET between two fluorescent phospholipids, we show that extrusion transiently decreases (within 30 seconds) the phospholipid coverage of LDs by two-fold (**New Figure 2D-F**, a copy is shown in our response to point #2 of the editor).

Other points:

P12-13 (and at other section of the manuscript): The authors frequently refer to phospholipid density. A consistent use of 'LD coverage with phospholipids' or 'phospholipid coverage' (as used by the authors) would be more suitable, clearly distinguishing it from lipid packing/lipid packing density at the molecular scale?

We agree that this terminology was misleading because we also use the term LD density when discussing the inclusion of brominated oil to increase their weight per volume. We now systematically use the term "coverage" to define the amount of phospholipid that covers the LDs.

Related to the abstract: The following section is not entirely clear and should be rephrased. 'Binding of purified PLIN3 and PLIN4 AH was dependent on tension, even with polar lipids favoring packing defects, and showed an inverse correlation between protein and phospholipid densities on LDs. In contrast, PLIN1 bound readily to LDs fully covered by phospholipids; PLIN2 showed an intermediate behavior.' Without further context and before reading the manuscript, it is not becoming sufficiently clear what is meant by 'even with polar lipids favoring packing defects' (why 'even'). Also, it is unclear which inverse correlation is meant. What does 'fully covered by phospholipids' mean in this context, and what is meant by intermediated behavior.

We have modified the abstract to improve its clarity.

Related to Figure 1C: Despite the text 'In striking contrast, PLIN4 displayed diffuse cytosolic signal in almost all non-treated cells at all stages of differentiation, with only faint staining of some peripheral LDs in a fraction of cells. Upon addition of OA, PLIN4 relocalized to peripheral LDs in a large fraction of cells (50% or more), with only a low number of cells retaining only cytosolic signal (Figure 1C).' I cannot find information on PLIN4 in Figure 1C. Have these data been represented in Figure 1D? Headlines to the individual panels might guiding the eye or the reader and supporting an immediate, intuitive understanding of the figure.

We apologize for this error in quoting Figure 1 panels. Indeed, this sentence referred to the data shown in Panels 1B (typical immunofluorescence images) and 1D

(quantification). Please note that we have moved one panel of the new Figure 1 in the supplementary Fig. S1 and, conversely, moved one panel from Fig S1 to Figure 1 (see your next suggestion).

Related to Figure 1/S1: It would help to move figure S1A forward in Figure 1. This would make Figure 1 immediately more 'accessible'.

We agree and have done this change.

Figure S1: The number of days of differentiation should be given in the Figure itself to improve the intuitive readability of the figure.

Done

p9 L22: 'High performance thin layer chromatography (HPTLC) (Fig. S2B) and SDS-PAGE (Figure 2B) showed that the majority ($\approx 75\%$) of lipids and of protein was recovered after extrusion.'. It is not entirely clear if this statement holds for all components equally: Proteins, phospholipids, and TAG. What is the % recovery of fluorescently labeled PE? Could those be provided with the Figure?

We observed similar loss ($\approx -25\%$) in PLIN3, in phospholipids and in TG; see table below. We have added these values in the corresponding figure (Fig S2A for protein recovery; Fig S2B for lipid recovery).

Lipid and protein recovery upon extrusion (%)

PLIN3	Rho	Triolein	POPC
74	79	72	73

Related to P11 I262-263: The interesting observation that DOG does not promote PLIN3 binding in this experimental setup should be discussed (here or in the discussion) in light of earlier observations by Skinner et al., 2009; Ben M'barek et al., 2017; Choi et al., 2023; Stribny and Schneiter, 2023, which are cited in the original manuscript. Can the seemingly divergent observations be reconciled?

Previous experiments performed with liposomes by others showed a clear effect of DAG, which we also observed here using a different proxy (NBD fluorescence). However, the amplitude of the DAG-promoting effect appeared very small as compared to the effect of tension, which was not tested before and which yield to a spectacular increase in PLIN3 binding. Therefore, our work is consistent with previous observations but uncovers another parameter, phospholipid coverage that turns out to be more important than lipid composition.

It should be also mentioned that the distribution of DAG in LDs is not a trivial issue because the chemistry and polarity of this lipid species is intermediate between neutral lipids, such a triolein, and polar lipids, such as phospholipids. We thus examined the distribution of a fluorescent DAG analog (TopFluor DG) in our system and also performed molecular dynamics simulations. We did not observe an enrichment of TopFluor DG at the surface of the aLD. Instead, TopFluor DG distributed in the LD core as well (see photo, below). It is possible that the fluorescent moiety changes the partitioning of the lipid. In molecular dynamics simulations that DAG tends to be more localized at the surface compared to triacylglycerol (TAG).

TopFluor DG distribution

This open issue might explain why the effect of DAG on PLIN3 recruitment to our aLDs is less striking compared to the findings by Skinner et al. (2009). In their study, the effect of DAG on PLIN3 recruitment was investigated in cellulo by inducing lipolysis and using a DAG lipase inhibitor to accumulate DAG on LDs. In this scenario, because lipolysis occurs at the surface of the LD, DAG may transiently accumulate at the surface, thereby promoting PLIN3 recruitment.

Finally, it should also be mentioned that the effect of DAG on LDs might not be restricted to the control of protein recruitment. Vanni and colleagues showed compelling simulations and experiments, which suggest that diacylglycerols facilitate LD budding from the ER by decreasing the critical concentration at which triolein starts de-mixing from the lipid bilayer (PMID: 33522484). This important paper is now quoted.

In conclusion, our results do not contradict those obtained by others. We do agree that DAG in bilayer impacts PLIN binding and, in addition, that it might facilitate LD emergence. However, the impact of DAG on the surface properties of isolated LDs is much lower than the impact of tension, which was not studied before.

Overall, a very interesting study overcoming a series of technical challenges! Very suitable for the readership of JCB.

Reviewer #2

The study conducted by Araujo et al. aims to delve into the mechanisms underlying the specific targeting of Plin 1 to 4 to distinct subsets of LDs, particularly in human adipocytes. Specifically, they seek to unravel how Plin1 predominantly localizes to large and mature LDs, typically located in the cell center, while Plin2-4 target smaller LDs primarily situated at the periphery, especially upon oleate loading. After revisiting a long-standing observation, Figure 1, the authors introduce a novel assay integrating FACS, floatation, and imaging techniques, as depicted in Figure 2. This innovative method involves droplet extrusion to enhance the surface-to-volume ratio, thereby increasing monolayer packing defects, and subsequently monitoring the recruitment of

purified Plin1-4. The study explores the influence of DPPC, POPC, and DOPC on protein recruitment, as well as the impact of LD size. While numerous research groups, including the authors' group, have already devoted tremendous efforts to elucidate the factors governing PLIN protein gradual binding, the techniques developed in this study are notably compelling, meticulously executed, and promising for a better understanding of PLIN membrane association. However, this reviewer feels a lack of clarity in connecting the findings across the figures. The authors suggest that differential LD tensions are responsible for the binding hierarchy of Plin1 >> Plin2 >> Plin3. This referee fails to see data supporting this notion. Nonetheless, this reviewer supports the publication of the study but strongly recommends that the authors consider addressing the following concerns, by order of priority. Thank you for these overall very positive comments.

1- The authors posit that LDs in differentiated hWA exhibit varying degrees of tension, which likely relates to lipid packing defects, with smaller LDs displaying a higher depletion of phospholipids compared to larger ones. This tension disparity is purported to influence the differential binding of Plin1-4 to LDs of varying sizes. Given this premise, it would be pertinent for the authors to quantify the level of packing defects in these LDs. One potential approach could involve utilizing packing defect sensors within cells; with Bruno Antony's group being the pioneer in developing and mastering such sensors, this could/should be doable. These sensors could potentially detect differences in phospholipid coverage between LDs, shedding light on the underlying mechanisms driving Plin1-4 binding. Alternatively, the authors could opt to purify LDs, followed by trypsinization or not, before exposing them to Plins. For instance, the binding of Plin3 to specific small purified LDs, if observed, would provide compelling evidence for the existence of tension discrepancies between LDs and further validate the authors' hypothesis.

We thank the reviewer for these suggestions. First, we would like to note that the binding of PLIN's does not seem to depend so much on LD size; our experiments at different stages of LD differentiation show PLIN4 binding to LDs upon OA induction that are not necessarily much smaller than PLIN1 LDs, for example early in differentiation (see fig S1A), and even later in differentiation (Fig1C), more peripheral LDs are of the same size as PLIN4 LDs formed after OA addition. The defining difference is therefore how these LDs are generated and also their positioning in the cell. Second, as other mature adipocyte cell lines, hTERT WA cells are refractory to transfection methods, preventing us from expressing lipid packing sensors. However, the revision now includes two types of assays that better address the question of LD tension in cells and in vitro. In cells, we now compare two ways of adding oleic acid: either as a single burst or in a stepwise manner over a long period (**New Figure 6**, a copy is shown in our response to point #1 of the editor). In vitro, we now quantify what is happening during extrusion by a FRET assay (**New Figure 2D-F**, a copy is shown in our response to point #2 of the editor). These two approaches strengthen our model where differences in LD surface tension contribute to PLIN sorting.

2- While the authors tested the POPC and DOG mixture in 2D, it is noted that LDs are not exclusively composed of POPC. Therefore, conducting experiments with adequate phospholipid mixtures that better reflect LD composition, along with DOG, would provide more comprehensive information on the impact of DOG. This assay is pivotal, as the authors conclude that PLIN3 does not bind spontaneously to LDs even when surrounded by a DOG-enriched lipid monolayer, while it binds avidly to LDs under

tension. However, this conclusion seems to contradict previous studies that have shown the localization of Plin3 to nascent LDs containing DOG (Khaddaj et al (PMID: 37465010), Choi et al (PMID: 37268630) and Chung et al (PMID: 31708432)).

We now present new LD binding experiments in which we used LDs covered by PC, PE and PI to mimic what has been reported on authentic LDs in two lipidomic studies (PMID: 22872753, PMID: 17210984). We performed these experiments in the presence or in the absence of DAG. The corresponding data are shown in a new Figure (Figure 3C, D; a copy is shown in our response to point #3 of the editor). They indicate that binding of PLIN3 remains highly dependent on the extrusion step, suggesting that tension, not lipid composition is the main factor that controls LD recruitment of PLIN3.

As for the apparent contradiction between our data and previous ones, please note we confirmed the effect of DAG in liposomes as seen by others using a different proxy (NBD fluorescence). Therefore, our data do not contradict previous investigations. However, the amplitude of the DAG-promoting effect appears just very small as compared to the effect of tension, which was not tested before and which yields to a very large increase in PLIN binding.

3- This referee is unable to understand the "replacement terminology" model by the authors and the Arfgap data appears irrelevant to the manuscript. An alternative understanding of the data is that Plin3 exhibits better binding to LDs with lower rhodamine signal due to the extrusion process leading to daughter LDs with varying densities of phospholipids. Therefore, Plin3 may have a preference for binding to LDs with reduced phospholipid coverage, rather than displacing phospholipids. The experiments with beads may not be directly comparable, as the lipid density is affected by methylation, allowing Plin3 to bind to beads with more space, such as the methylated phospholipid beads. Further clarification and detail are needed to support the notion that Plin3 displaces phospholipids, as proposed in the experiments and rationale provided.

3a. We agree with the reviewer that Plin3 has a preference for binding to LDs with reduced phospholipid coverage. The new FRET experiments in Figure 2D-F (a copy is shown in our response to point #2 of the editor) strengthen this hypothesis by showing that extrusion causes the transient spreading of the phospholipid monolayer, which relaxes back to normal within 30 seconds. In the presence of PLIN3, the recovery amplitude was much smaller. Together with the fact that the decrease of Rhodamine signal upon extrusion is larger in the presence of Plin3 than in the absence of protein (see new Figure 5B), these experiments favor a model in which Plin3 binds in an opportunistic manner to LDs with low phospholipid coverage. We have clarified this point in our text, notably by removing the word 'displacement' and the verb 'displace'.

New quantification in Figure 5

3b. In contrast, we respectfully disagree with the reviewer who questions the importance of the data on Arf1-GTP (not ArfGAP, which was not studied here). These experiments are very relevant because Arf1-GTP can be found associated with different surfaces in the cells, both membrane-bound organelles and lipid droplets, whereas PLIN3 is much more selective to lipid droplets. Because both Arf1-GTP and PLIN3 use amphipathic helices (AH), comparing their LD binding properties in vitro was very informative. We show Arf1-GTP and PLIN3 display very different sensitivity to LD tension. As previously shown for membrane curvature sensors (see e.g. PMID: 19927117), all AHs are not the same and it is critical to distinguish those that act as sensors of lipid surfaces and those that are strong binders. In the present study, the comparison between PLIN3 and Arf1GTP shows that AH-containing proteins are very different in their response to LD surface tension.

3c. We agree that diphytanoyl lipids are special. However, the very fact that the fluorescence of Rhodamine-PE in diphytanoyl-based bilayers is not affected at all by the addition of PLIN3 suggests that the decrease in Rhodamine-PE that we observed when PLIN3 binds to LDs is due to the specific arrangement of lipids in this context (notably interdigitation). This is also supported by our molecular dynamic simulations.

4- The use of brominated vegetable oil at a concentration of 33% represents a substantial proportion. Given that bromine and hydrogen have markedly different solubility in water a priori, there is a potential discrepancy in the molecular architecture of the brominated oil at the LD interface compared to the triacylglycerols utilized in this study. Although the authors acknowledged this issue in figure S2, the impact of bromine before extrusion appears evident, particularly in the 10 μ m-sized LDs as observed figure S2D. To address this point, the referee encouraged the authors to conduct simple experiments comparing the use of 100% brominated oil versus 100% triacylglycerols in aLDs. Demonstrating that the presence of bromine does not significantly influence the results would provide crucial validation for the experimental findings.

The revision now shows a new experiment in which we replaced triolein by brominated oil in a gradual manner (0, 33, 66 and 100%). Binding of Plin3 under these four conditions remained conditioned by extrusion (new supplementary FigS2. A copy of this figure is shown in our response to point #4 of the editor).

5- A key aspect is missing from the manuscript, which is the range of tension alteration induced by the extrusion process. It may be extremely useful to quantify the extent of tension change when one droplet is divided into two through extrusion for example. Providing quantitative values for tension alteration would enhance understanding the experimental setup and its implications. Additionally, the authors could compare their values with those recently measured by the Ditscher group (PMID: 37212777) to provide further context and validation. Including such quantitative data would strengthen the rigor and interpretation of the findings.

We agree. To address this point, i.e., to estimate the impact of extrusion on the coverage of the LDs by phospholipids, we performed FRET experiments, which are presented in the new **Figure 2 (panels D to F)**, a copy of this figure is shown in our response to point #2 of the editor). We assessed the spreading of the phospholipid monolayer by measuring FRET between NBD-PE and Rho-PE, a pair that is classically

used to monitor bilayer fusion. Upon extrusion, we recorded a decrease in FRET, which recovers with a fast kinetic (half time \approx 10 seconds). Adding unlabeled PLIN3 reduces FRET recovery. These measurements validate our hypothesis and method and suggest that the spreading of the phospholipid monolayer upon extrusion is transient - in the range of two-fold - and is stabilized by the insertion of PLIN3. These experiments give a good view of what is happening during extrusion and strengthen the rationale of our assay.

Minor.

- Line 404 "At high PC coverage density (i.e. low tension), the surface of the PC monolayer was comparable to the surface of a bilayer, showing an increase in lipid packing defects according to the level of PC unsaturation (14:0/14:0 < 16:0/18:1 < 18:1/18:1)."

The authors need to provide clearer clarification regarding the contribution of their simulation study in comparison to the existing work by Farese/Walther, Vanni, and Swanson groups. Specifically, they should delineate how their simulation adds value or insight beyond what has already been explored by these groups. This clarification will help readers understand the unique perspective or findings offered by the authors' simulation with the existing body of literature.

In previous simulations, lipid unsaturation and lipid coverage were studied independently (Vanni studied tension, Swanson tested unsaturation). Our simulations combine the two parameters. This combination allows us to demonstrate that phospholipid coverage has a much stronger effect on the surface properties of LDs than the ratio between saturated and monounsaturated acyl chains in phospholipids.

- Line 414 "The LD surface packing properties depend mostly on the oil reservoir underneath, an effect that becomes prominent at low phospholipid density."

This is a sufficient rationale for conducting experiments specifically testing the impact of brominated lipids at 100% concentration.

Done, see above

- Please clarify whether the protein concentration was in excess relative to the available surface area of the aLDs

Thank you for this remark. The protein and the phospholipid concentrations were chosen to be compatible with the available LD surface area. For the experiments conducted with small (2 μ m) LDs, the actual phospholipid concentration was 62.5 μ M. An amphipathic helix of \approx 100 aa has a length of $0.15 \times 100 = 15$ nm and a width of about 1 nm, hence a surface of ≈ 15 nm². The average surface of a phospholipid is in the range of 0.7 nm². Therefore, a 100-aa helix occupies a surface comparable to 20 phospholipids. To have the helix and the protein on equal footing in term of potential LD coverage, we should thus use $62.5 / 20 \approx 3$ μ M protein. In effect, we used 1-2 μ M (hence a \approx 1:50 mol:mol ratio) to also take into account the fact other regions besides the central 11-mer repeats of perilipins might contribute to binding or at least occupy a surface above the LD surface. The same reasoning applies to the LDs used for light microscopy observations: here the phospholipid concentration was 12.5 μ M and the

protein concentration was 0.25 μM , hence a 1:50 mol:mol ratio. The new material and method section now includes a paragraph on this point.

141 "a fraction of cells. Upon addition of OA, PLIN4 relocalized to peripheral LDs in a large fraction 142 of cells (50% or more), with only a low number of cells retaining only cytosolic signal (Figure 1C)." Should be 1D.

Correction done.

Figure 1D needs improvement as it is currently difficult to comprehend. We reorganized and clarified the categories in this panel. We also removed from the former panel 1C, which was distracting (now shown in Fig S1).

Reviewer #3:

This manuscript by Araujo et al investigates the targeting of perilipin proteins to lipid droplets (LDs), predominantly using a novel in vitro approach. Major conclusions are that PLIN1 behaves very different from other perilipins, and that extrusion as a model for a temporary increase in surface tension due to deformation of in vitro formed LDs allows some perilipins to bind LDs that otherwise would not bind LDs. Overall, the manuscript reports well done and well controlled experiments, is well written (with the exception of referencing) and reports results for an interesting area of cell biology.

Weaknesses in my opinion are that:

i) extrusion is a hard to control process with somewhat unclear consequences. In the revision, we present a new strategy to quantify the impact of extrusion on the surface properties of LDs (Figure 2D-F). See details in our answer to your point #2.

ii) the main result, PLIN1 behaves different than the other PLINs, appears a bit trivial since the protein does not only have amphipathic helices used by other PLINs to bind LDs but also large, membrane embedded stretches of amino acid sequence. This might sound trivial. However, our study is the first that effectively compares full-length PLIN1, PLIN2 and PLIN3 in a reconstitution system, which allows us to show that their properties are drastically different.

iii) a key aspect of emerging model (PLIN binding is responsive to the speed of LD expansion) is not tested in cells.

We now show cellular experiments to test this hypothesis (new Figure 6) and comments below (response to your point #1)

Overall, this is an interesting and well performed manuscript with some limitations that could be better discussed. The main points, I would recommend to address are:

1) Can the authors test whether varying the speed of LD formation (e.g., varying oleate concentrations and measuring LD growth and TG synthesis, using inhibitors of TG synthesis as controls added at different time points) affects PLIN targeting in cells? To me this would be important (and possible quite simple?) to add.

Thank you for this suggestion, which prompted us to perform new experiments using TERT-hWA cells. Specifically, we compared the subcellular distribution of endogenous

perilipins by immunofluorescence when we challenged the cells with a burst of oleic acid (100 μ M) or when added the same amount of fatty acid in a stepwise manner (5 x 20 μ M). The results, which are shown in the **new Figure 6** (a copy is shown in our response to point #1 of the editor), indicate that PLIN4 preferentially binds to LDs under the first condition. This experiment provides cellular evidence for our model; namely that PLIN3/4 are adapted to LD binding under tension conditions.

2) While the conclusions on the extrusion experiments are very reasonable, I wonder whether an orthogonal method (e.g. with a drop tensiometer) could be used to control surface tension and assay binding?

We don't have access to a drop tensiometer. However, we now better address the effect of extrusion on the surface properties of LDs by performing FRET experiments between fluorescent lipids at the LD surface. These experiments (new **Figure 2D-F**, a copy is shown in our response to point #2 of the editor) show that extrusion causes transient spreading of the LD surface by two-fold, which recovers with a half-time of \approx 10 seconds. With PLIN3 present, we observed much less recovery, suggesting that PLIN3 takes advantage of the transient surface extension of the LDs to bind to them.

3) References in the manuscript seem a bit random and often appear to not well consider some of the older, original literature. The authors might want to carefully check this.

We now include the following references in the list of the first papers that describe the perilipin and their association with LDs.

Greenberg AS, Egan JJ, Wek SA, Garty NB, Blanchette-Mackie EJ, Londos C. Perilipin, a major hormonally regulated adipocyte-specific phosphoprotein associated with the periphery of lipid storage droplets. *J Biol Chem*. 1991 Jun 15;266(17):11341-6. PMID: 2040638.

Brasaemle DL, Barber T, Wolins NE, Serrero G, Blanchette-Mackie EJ, Londos C. Adipose differentiation-related protein is an ubiquitously expressed lipid storage droplet-associated protein. *J Lipid Res*. 1997 Nov;38(11):2249-63. PMID: 9392423.

Blanchette-Mackie EJ, Dwyer NK, Barber T, Coxey RA, Takeda T, Rondinone CM, Theodorakis JL, Greenberg AS, Londos C. Perilipin is located on the surface layer of intracellular lipid droplets in adipocytes. *J Lipid Res*. 1995 Jun;36(6):1211-26. PMID: 7665999.

Scherer, P.E., P.E. Bickel, M. Kotler, M. and H.F. Lodish. 1998. Cloning of cell-specific secreted and surface proteins by subtractive antibody screening. *Nat Biotechnol* 16:581–586. doi: 10.1038/nbt0698-581.

We also quote the following paper as a new example of efforts to make artificial LDs:

Gandhi SA, Parveen S, Alduhailan M, Tripathi R, Junedi N, Saqallah M, Sanders MA, Hoffmann PM, Truex K, Granneman JG, Kelly CV. Methods for making and observing model lipid droplets. *Cell Rep Methods*. 2024 May 20;4(5):100774. doi: 10.1016/j.crmeth.2024.100774. Epub 2024 May 14. PMID: 38749444.

This paper as an example of the diversity in protein coverage of LDs:

Thul PJ, Tschapalda K, Kolkhof P, Thiam AR, Oberer M, Beller M. Targeting of the *Drosophila* protein CG2254/Ldsdh1 to a subset of lipid droplets. *J Cell Sci.* 2017 Sep 15;130(18):3141-3157. doi: 10.1242/jcs.199661. Epub 2017 Aug 3. PMID: 28775149.

Finally, we quote this paper for the complex effect of lipid composition on LD formation:

Zoni V, Khaddaj R, Campomanes P, Thiam AR, Schneiter R, Vanni S. Pre-existing bilayer stresses modulate triglyceride accumulation in the ER versus lipid droplets. *Elife.* 2021 Feb 1;10:e62886. doi: 10.7554/eLife.62886.

4) A procedural, but important point: all reported results in the paper need quantitation (absent of most of the cell biology experiments) and appropriate statistics (which seems not reported for any of the results); for an example lacking either, see figure 1. This is essential in my opinion.

In Figure 1, a quantification is provided for one representative experiment showing three different days of differentiation, and three independent quantifications of the distribution of PLIN4 for day 10 are now shown in new Figure 6, illustrating that the effect of oleic acid addition on PLIN4 distribution is reproducible. We also acknowledge the variability of these experiments, which are due to the complex nature of this biological sample. We therefore prefer to show individual experiments.

In the revision, we have also increased the number of independent experiments in Figures 5 and S2B and included statistical tests (Figures 2B, 2C, 3B, 3C 4C, 4E, 5B). In addition, we not only specify the number of independent experiments but also show the individual values. When suitable, we show the data as super-plots, which is the best way to capture the variability of measurements in a fair and comprehensive manner (the super-plots were first introduced in a *J Cell Biol* paper; Lord et al. *J Cell Biol.* 2020 PMID: 32346721).

Finally, it should be noticed that the robustness of our work relies on the fact that we use different approaches (flotation, FACS, and microscopy observations of cellular and artificial LDs), which all concur with the same conclusion: tension is a decisive factor for controlling protein targeting to LDs.

August 14, 2024

RE: JCB Manuscript #202403064R

Dr. Bruno Antony
Institut de Pharmacologie Moléculaire et Cellulaire
Institut de Pharmacologie Moléculaire et Cellulaire
Université Côte d'Azur, CNRS, Inserm, UMR7275
660 route des lucioles
Valbonne 06560
France

Dear Dr. Antony:

Thank you for submitting your revised manuscript entitled "Surface tension-driven sorting of human perilipins on lipid droplets". We have assessed your thorough revisions and would be happy to publish your paper in JCB pending final revisions necessary to meet our formatting guidelines (see details below).

A. MANUSCRIPT ORGANIZATION AND FORMATTING:

- 1) Text limits: Character count for Articles is < 40,000, not including spaces. Count includes abstract, introduction, results, discussion, and acknowledgments. Count does not include title page, figure legends, materials and methods, references, tables, or supplemental legends.
- 2) Figures limits: Articles may have up to 10 main text figures.
- 3) Figure formatting: Scale bars must be present on all microscopy images, including inset magnifications. Molecular weight or nucleic acid size markers must be included on all gel electrophoresis. In order to accommodate readers with red-green color blindness, we suggest that you change all red/green color schemes.
- 4) Statistical analysis: Error bars on graphic representations of numerical data must be clearly described in the figure legend. The number of independent data points (n) represented in a graph must be indicated in the legend. Statistical methods should be explained in full in the materials and methods. For figures presenting pooled data the statistical measure should be defined in the figure legends. Please also be sure to indicate the statistical tests used in each of your experiments (either in the figure legend itself or in a separate methods section) as well as the parameters of the test (for example, if you ran a t-test, please indicate if it was one- or two-sided, etc.). Also, if you used parametric tests, please indicate if the data distribution was tested for normality (and if so, how). If not, you must state something to the effect that "Data distribution was assumed to be normal but this was not formally tested."
- 5) Abstract and title: The abstract should be no longer than 160 words and should communicate the significance of the paper for a general audience. The title should be less than 100 characters including spaces. Make the title concise but accessible to a general readership.
- 6) Materials and methods: Should be comprehensive and not simply reference a previous publication for details on how an experiment was performed. Please provide full descriptions in the text for readers who may not have access to referenced manuscripts.
- 7) All antibodies, cell lines, animals, and tools used in the manuscript should be described in full, including accession numbers for materials available in a public repository such as the Resource Identification Portal. Please be sure to provide the sequences for all of your primers/oligos and RNAi constructs in the materials and methods. You must also indicate in the methods the source, species, and catalog numbers (where appropriate) for all of your antibodies. Please also indicate the acquisition and quantification methods for immunoblotting/western blots.
- 8) Microscope image acquisition: The following information must be provided about the acquisition and processing of images:
 - a. Make and model of microscope
 - b. Type, magnification, and numerical aperture of the objective lenses

- c. Temperature
- d. Imaging medium
- e. Fluorochromes
- f. Camera make and model
- g. Acquisition software
- h. Any software used for image processing subsequent to data acquisition. Please include details and types of operations involved (e.g., type of deconvolution, 3D reconstitutions, surface or volume rendering, gamma adjustments, etc.).

10) Supplemental materials: There are strict limits on the allowable amount of supplemental data. Articles may have up to 5 supplemental figures. Please also note that tables, like figures, should be provided as individual, editable files. A summary of all supplemental material should appear at the end of the Materials and methods section.

13) ORCID IDs: ORCID IDs are unique identifiers allowing researchers to create a record of their various scholarly contributions in a single place. Please note that ORCID IDs are now *required* for all authors. At resubmission of your final files, please be sure to provide your ORCID ID and those of all co-authors.

Please note that JCB now requires authors to submit Source Data used to generate figures containing gels and Western blots with all revised manuscripts. This Source Data consists of fully uncropped and unprocessed images for each gel/blot displayed in the main and supplemental figures. Since your paper includes cropped gel and/or blot images, please be sure to provide one Source Data file for each figure that contains gels and/or blots along with your revised manuscript files. File names for Source Data figures should be alphanumeric without any spaces or special characters (i.e., SourceDataF#, where F# refers to the associated main figure number or SourceDataFS# for those associated with Supplementary figures). The lanes of the gels/blots should be labeled as they are in the associated figure, the place where cropping was applied should be marked (with a box), and molecular weight/size standards should be labeled wherever possible. Source Data files will be made available to reviewers during evaluation of revised manuscripts and, if your paper is eventually published in JCB, the files will be directly linked to specific figures in the published article.

Journal of Cell Biology now requires a data availability statement for all research article submissions. These statements will be published in the article directly above the Acknowledgments. The statement should address all data underlying the research presented in the manuscript. Please visit the JCB instructions for authors for guidelines and examples of statements at (<https://rupress.org/jcb/pages/editorial-policies#data-availability-statement>).

B. FINAL FILES:

-- Cover images: If you have any striking images related to this story, we would be happy to consider them for inclusion on the journal cover. Submitted images may also be chosen for highlighting on the journal table of contents or JCB homepage carousel.

Images should be uploaded as TIFF or EPS files and must be at least 300 dpi resolution.

****It is JCB policy that if requested, original data images must be made available to the editors. Failure to provide original images upon request will result in unavoidable delays in publication. Please ensure that you have access to all original data images prior to final submission.****

****The license to publish form must be signed before your manuscript can be sent to production. A link to the electronic license to publish form will be sent to the corresponding author only. Please take a moment to check your funder requirements before choosing the appropriate license.****

Thank you for your attention to these final processing requirements. Please revise and format the manuscript and upload materials within 7 days. If you need an extension for whatever reason, please let us know and we can work with you to determine a suitable revision period.

Thank you for this interesting contribution, we look forward to publishing your paper in Journal of Cell Biology.

Sincerely,

William Prinz, PhD
Monitoring Editor

Andrea L. Marat, PhD
Deputy Editor

Journal of Cell Biology